# Deciphering of SOX9 Functions in Pancreatic Cancer Cells

**DOI:** 10.3390/ijms26062652

**Published:** 2025-03-15

**Authors:** Kirill Kashkin, Liya Kondratyeva, Eugene Kopantzev, Ivan Abramov, Lyudmila Zhukova, Igor Chernov

**Affiliations:** 1Shemyakin-Ovchinnikov Institute of Bioorganic Chemistry of the Russian Academy of Sciences, Miklukho-Maklaya, 16/10, 117997 Moscow, Russia; kopantzev@gmail.com (E.K.); igor.palich@gmail.com (I.C.); 2GBUZ Moscow Clinical Scientific and Practical Center Named After A.S. Loginov MHD (MCSC), 111123 Moscow, Russia; i.abramov@mknc.ru (I.A.); l.zhukova@mknc.ru (L.Z.)

**Keywords:** pancreatic cancer, SOX9, knockdown, siRNA, pathway analysis, gene signature

## Abstract

SOX9 is widely regarded as a key master regulator of gene transcription, responsible for the development and differentiation programs within tissue and organogenesis, particularly in the pancreas. SOX9 overexpression has been observed in multiple tumor types, including pancreatic cancer, and is discussed as a prognostic marker. In order to gain a more profound understanding of the role of SOX9 in pancreatic cancer, we have performed *SOX9* knockdown in the COLO357 and PANC-1 cells using RNA interference, followed by full-transcriptome analysis of the siRNA-transfected cells. The molecular pathway enrichment analysis between *SOX9*-specific siRNA-transfected cells and control cells reveals the activation of processes associated with cellular signaling, cell differentiation, transcription, and methylation, alongside the suppression of genes involved in various stages of the cell cycle and apoptosis, upon the *SOX9* knockdown. Alterations of the expression of transcription factors, epithelial–mesenchymal transition markers, oncogenes, tumor suppressor genes, and drug resistance-related genes upon *SOX9* knockdown in comparison of primary and metastatic pancreatic cancer cells are discovered. The expression levels of genes comprising prognostic signatures for pancreatic cancer were also evaluated following *SOX9* knockdown. Additional studies are needed to assess the properties and prognostic significance of SOX9 in pancreatic cancer using other biological models.

## 1. Introduction

The alterations in the expression levels of key regulatory factors in cancer cells have attracted a significant research interest, both for purpose of detailed analysis of their functions and as potential targets for tumor therapy. In particular, the transcription factor SOX9 is of great interest. SOX9 belongs to group SOXE of determining region Y (SRY)-related high-mobility group (HMG) box (SOX) proteins. SOX9, as well as other SOX proteins, possess a transactivation domain that is responsible for transcription regulation and is unique for SOXE proteins. SOX9, as well as other SOXE proteins, SOX8 and SOX10, contain a dimerization domain located proximally to HMG box [1]. The dimerization domain ensures the formation of homologous and heterologous dimers between SOXE proteins but not between SOXE and other SOX proteins [2].

SOX9 has been identified as a crucial transcriptional regulator of various developmental processes including bone formation, chondrogenesis, the development of the testis, the nervous tissue, the mammary gland, and the pancreas [3,4].

SOX9 is involved in numerous signaling pathways that are crucial for organogenesis, epithelial–mesenchymal transition (EMT), and cancer progression. These pathways encompass, but are not limited to, transforming growth factor-β (TGF-β) signaling, C-jun/SMAD, Wnt/β-catenin, Sonic Hedgehog (SHH), and mitogen-activated protein kinase (MAPK)/extracellular signal-regulated kinase (ERK). The dysregulation of pathways such as nuclear factor kappa B (NF-κB), epidermal growth factor (EGFR), and neurogenic locus notch homolog protein 1 (Notch1) also has been demonstrated to transcriptionally activate SOX9 [1,5]. In some pathways, SOX9 has complicated and interconvertible regulation as in case of Wnt/β-catenin pathway, where SOX9 may play roles of upstream or downstream effector, being the activator or repressor of the pathway [6].

SOX9, as well as SOX8 and SOX10, cooperate directly with each other and indirectly with other SOX transcription factors through numerous signaling pathways. Specifically, SOX9 and SOX10 have been observed to regulate the Wnt/β-catenin pathway in a cooperative manner, with SOX9 having a multifaceted role in the regulation. At the same time, SOX2 and SOX4 have been implicated in activating the Wnt/β-catenin pathway, while SOX1, SOX15, and SOX17 have been identified as suppressors. SOX2 and SOX4 have been associated with the TGF-β pathway, which has been shown to stimulate SOX9 expression. Additionally, SOX2, SOX9, and SOX10 have been identified as participants of Notch1 signaling pathway, playing a crucial role in sustaining cancer stem cells and promoting metastasis. Furthermore, SOX4 and SOX9 have been implicated in the promotion of epithelial–mesenchymal transition (EMT) and metastasis via the TGF-β pathway, with SOX7 having the capacity to counteract this effect [1,5].

The inflammatory environment has been shown to extensively interact with protumor and cancer stem cells, thereby promoting tumor initiation and development. A growing body of research has identified a connection between chronic inflammation and various cancer types, including pancreatic cancer [7,8]. SOX2, SOX9, and SOX10 have been implicated in several pathways crucial to chronic inflammation, such as NF-κB, MAPK/ERK, TGF-β pathways, and cytokine networks [5,9]. SOX9 up-regulation by pro-inflammatory interleukin 1 beta (IL-1β) may be associated with inflammation [10], and the inactivation of SOX9 was shown to reduce inflammation [11]. These data provide a rationale for considering SOX9 as a potential target for therapy for both chronic inflammation itself and inflammation associated with cancer progression, including pancreatic cancer.

The deregulation of *SOX9* expression has been implicated in carcinogenesis in different organs and tissues. *SOX9* is one of the most important genes up-regulated during early tumor formation in breast, lung cancer, hepatocarcinoma cells, esophageal squamous cell carcinoma cells, osteosarcoma, and pancreatic cancer. Up-regulated *SOX9* expression plays a role in acquiring stem cell phenotypes by cancer precursor cells and in initiating cancer-associated EMT process during tumor progression. The notable effect of *SOX9* overexpression on cell motility appears to be associated with the weakening of tight junctions [1]. Inhibition of SOX9 has been observed to result in the decrease in cell growth and tumorigenic properties in glioma, renal cancer, and papillary thyroid cancer [1,5]. Conversely, SOX9 itself has been shown to play a suppressor role in cervical cancer [12], endometrial carcinoma [13], melanoma [14], and certain intestinal tumors [15]. In the context of pancreatic cancer cells, the inhibition of *SOX9* has been shown to induce cell-line-specific effects on proliferative activity, expression of EMT markers, and activation of apoptotic caspases [16].

Differential expression of *SOX9* has been positively correlated with drug resistance of several types of tumors [17]. In particular, pancreatic cancer cells with high levels of SOX9 exhibit increased resistance to gemcitabine in comparison to cells with low levels of this protein. This phenomenon may be attributed to the induction of putative cancer stem cell characteristics [18]. Consequently, SOX9 is regarded as a prognostic marker of pancreatic cancer progression and treatment [19].

In vitro and in vivo experiments have revealed that SOX9 plays a pivotal role in promoting proliferation, progression, and metastasis of tumor cells, including pancreatic adenocarcinoma (PAAD) cells, by maintaining their stemness and facilitating epithelial–mesenchymal transition [16,20]. Our studies have demonstrated that the suppression of *SOX9* by small interfering RNAs (siRNAs) in PAAD cells leads to a reduction in cell proliferation, an inhibition of cell motility and migration, and the induction of apoptosis. However, the impact of SOX9 on the expression of individual cell cycle regulator proteins and markers of the epithelial–mesenchymal phenotype remained ambiguous and contingent upon the specific cell line. The down-regulation of the *SOX9* expression did not result in the anticipated enhancement of the epithelial status of the pancreatic cancer cell lines [16]. While the role of SOX9 in carcinogenesis is evidently significant, the mechanisms underlying its functions remain to be fully elucidated.

In order to study the role of SOX9 in pancreatic cancer further, we have conducted full-transcriptome analysis of cells from two PAAD cell lines, PANC-1 and COLO357, under conditions of suppression of *SOX9* gene expression using siRNAs. In the present work, we were first to compare full transcriptomes of these cell lines under *SOX9* knockdown taking into account the origin, and morphological and genetic peculiarities of the cells.

The PANC-1 cells originated from primary pancreatic ductal adenocarcinoma [21], while COLO357 cells were isolated from a metastasis of pancreatic ductal adenocarcinoma to lymph node [22]. The fact that both patients were not treated with antitumor therapy prior to the cell lines establishment is of particular importance. According to a number of morphological and growth characteristics, PANC-1 cells belong to the low-differentiated (G3) subtype, and COLO357 cells to the intermediate or mixed subtype (G1–G3) [23].

According to Collison et al. [24], COLO357 and PANC-1 cells belong to the quasi-mesenchymal (QM-PDA) cell subtype. In contrast to the classical subtype (classical PDA), QM-PDA cells are characterized by high expression of mesenchymal genes.

COLO357 and PANC-1 cells contain a large number of chromosomal abnormalities and mutations. In particular, the KRAS oncogene (p.Gly12Asp) is mutated in both cell lines and the CDKN2A/p16 gene is damaged (p.Met52Lys mutation in COLO357 and homozygous deletion in PANC-1). Additionally, the p53 gene is mutated in PANC-1 (p.Arg273His), while COLO357 contain the wild-type p53 gene [25,26].

The objective of the present study was to enhance comprehension of the functions of SOX9 in the processes of proliferation, epithelial–mesenchymal transition, metastasis, and other processes that are significant for the development of PAAD. To this end, we have conducted a full-transcriptome analysis of pancreatic cancer cell lines under *SOX9* knockdown. The obtained RNA-seq results were analyzed through a number of current biological process databases, including Gene Ontology, Reactome, and GSEA, and compared with specific and prognostic gene signatures of PAAD.

## 2. Results

### 2.1. Generation and Evaluation of cDNA Libraries

We performed SOX9 knockdown in PANC-1 and COLO357 cells by transient transfection of the cells with siRNA to *SOX9*, which resulted in almost 20-fold suppression of SOX9 protein product expression relative to corresponding cells transfected with neutral siRNA, siNEG (Figure 1a).

Following the confirmation of the *SOX9* knockdown at the protein level, the preparation of cDNA libraries was performed. Twelve PCR-amplified cDNA libraries from four groups of samples were obtained: PANC-1 and COLO357 pancreatic cancer cell lines with siRNA-suppressed *SOX9* gene expression (PANC-1–siSOX9, COLO357–siSOX9), and corresponding control cells transfected with neutral siRNA (PANC-1–siNEG, COLO357–siNEG), with three biological repeats for each group.

The prepared cDNA libraries were sequenced using Illumina NovaSeq 6000 technology, resulting in the acquisition of over 255 million (255,289,641) reads, with an average length of 100 bp. The average number of reads per library was 21.3 million (Appendix A).

The total number of genes for which significant differential expression data were obtained (*p*-adj < 0.05; TPM > 5) was 6010 genes for PANC-1 and 2354 genes for COLO357 (Appendix A).

*SOX9* gene knockdown in PANC-1 cells resulted in a more than 1.5-fold change in the expression levels of 2410 genes, of which 1327 genes were down-regulated and 1083 were up-regulated relative to siNEG RNA-transfected cells (Figure 1b). Knockdown of the *SOX9* gene in the COLO357 cell line resulted in a more than 1.5-fold change in the expression of 487 genes, of which 364 genes were down-regulated and 123 genes were up-regulated relative to control cells (Figure 1c).

A comparison of the two cell lines reveals the presence of common differentially expressed genes, including 162 genes with decreased expression and 46 genes with increased expression under *SOX9* knockdown, relative to the corresponding siNEG cells. To validate the obtained RNA-seq results, we have selectively determined the expression of a number of genes with different expression levels in cells transfected with neutral and *SOX9*-specific siRNAs using real-time PCR and Western blotting of cell lysates with antibodies to the corresponding proteins [16]. A comparison of the results shows high similarity between the changes in gene expression levels obtained by the three methods.

### 2.2. The Transcriptomic Alterations of PAAD Cells upon Knockdown of the SOX9 Gene

To investigate the effect of SOX9 knockdown on cell transcriptomes, we compared gene expression levels in cells transfected with SOX9-specific siRNA (COLO357–siSOX9, PANC-1–siSOX9) and in corresponding cells transfected with neutral siRNA (hereafter referred to as control cells, or COLO357–siNEG, PANC-1–siNEG).

#### 2.2.1. Gene Ontology

In order to ascertain the effect of *SOX9* knockdown in PANC-1 and COLO357 cells, GeneOntology.org/Panther ver.18.0 service was utilized, employing Fisher test against all human genes with Bonferroni *p*-correction.

Among the “Biological process” annotations, the GO therm “regulation of signal transduction” was identified for up-regulated upon *SOX9* knockdown genes in both cell lines under study. In case of down-regulated genes, the common “Molecular function” annotation “protein binding” was found, as well as the common “Cellular components” annotations “endomembrane system” and “cytoplasm”. Panther Pathway analysis revealed increased expression of “Gonadotropin-releasing hormone receptor pathway” genes in COLO357 and PANC-1 cells upon *SOX9* knockdown relative to corresponding control cells (Table 1, Appendix A).

At the same time, significant differences were identified between COLO357 and PANC-1 cells with respect to the impact of *SOX9* knockdown. Consequently, in COLO357–siSOX9 cells, processes associated with “response to endogenous stimuli” were activated while “protein processing” and “regulation of cell adhesion” were suppressed relative to control cells (Figure 2a). In PANC-1 cells with SOX9 knockdown, the processes related to development, differentiation, and positive regulation of transcription were most activated, while DNA replication, cell cycle regulation, and cellular response to stress were suppressed (Figure 2b).

The analysis of “Molecular function” according to GO in COLO357–siSOX9 cells reveals the suppression of serine-type peptidases and serine hydrolases, as well as protein binding, relative to control cells. In PANC-1 cells, *SOX9* knockdown resulted in the activation of transcription regulators and transcription-associated protein binding processes of regulatory DNA sequences; however, the binding of individual nucleotides and cytoskeleton proteins was suppressed (Appendix A).

The analysis of “Cellular component” according to GO revealed the reduction in the expression of gene-related extracellular membrane structures in COLO357–siSOX9 cells. In the case of PANC-1 cells, *SOX9* knockdown activated the expression of genes related to exocytosis, synaptic and transport vesicle membranes, and vacuole membranes, while genes related to contractile actin filaments, focal adhesion, and microtubules were suppressed. Among the genes suppressed by *SOX9* knockdown in PANC-1 cells, the annotations “cellular response to stress” (GO:0033554) and “detection of chemical stimulus” (GO:0009593) were noted also, which may be relevant to cellular response to drugs.

Expression of genes related to the Ras pathway (Panther pathway, P04393) was suppressed in PANC-1–siSOX9 cells relative to control cells (Appendix A).

#### 2.2.2. Reactome Gene Representation Analysis

The alterations in biochemical pathways in cancer cells upon suppression of SOX9 expression was investigated based on the expression levels of transcripts involved in each pathway using the Reactome online service (Pathway browser 3.7, database release 87, No interactors). Since the FDR value exceeded 25% in all cases of Reactome gene overrepresentation analysis, we present the pathways for which Entities *p* < 0.05.

In case of COLO357 cells, 36 pathways involving genes activated by SOX9 knockdown and 51 pathways involving genes suppressed expression under SOX9 knockdown (Entities *p* < 0.05) relative to cells transfected by neutral siRNA were revealed; in PANC-1 cells, the number of altered pathways was 43 and 81, respectively (Appendix A). The pathways altered in the same direction under SOX9 knockdown in both cell lines were also revealed: six pathways involving activated genes and seven pathways involving suppressed ones (Table 2).

Several processes exhibited divergent responses in COLO357 and PANC-1 cells upon SOX9 knockdown. In COLO357–siSOX9 cells, genes related to senescence-associated secretory phenotype (SASP) (R-HSA-2559582), RHO GTPases activate PKNs (R-HSA-5625740), and cellular senescence (R-HSA-2559583) processes were activated, while in PANC-1–siSOX9 cells, these processes were determined to be suppressed relative to corresponding control cells (Table 3).

Genes with increased and decreased expression were identified in COLO357–siSOX9 cells for the processes of interleukin 10 signaling (R-HSA-6783783), signaling by interleukins (R-HSA-449147), and ephrin signaling (R-HSA-3928664) (Table 4).

We have additionally investigated the most significantly different biochemical processes occurring in PAAD cells upon suppression of *SOX9* expression (Figure 3, Appendix A). The analysis of COLO357–siSOX9 cells shows 29 activated and 45 suppressed processes specific to this cell line (Entities *p* < 0.05). The analysis of PANC-1–siSOX9 cells reveals 35 activated and 74 suppressed processes not detected in COLO357–siSOX9 cells. Apparently, these differences are due to individual properties of the investigated cell lines.

#### 2.2.3. Reactome Gene Expression Analysis

The Reactome expression online service, PADOG, was also used to identify the processes altered in COLO357 and PANC-1 cells upon *SOX9* gene knockdown. The TPM values were used as arguments. Among the processes altered in *SOX9* knockdown cells, we found processes altered both equally and in opposite directions in the cells of the two cell lines under study (Table 5; Appendix A). The processes whose changes are most significant in COLO357–siSOX9 and PANC-1–siSOX9 cells were also revealed (Figure 4, Appendix A).

**Table 5 ijms-26-02652-t005:** Pathways changed in both COLO357 and PANC-1 cells with *SOX9* knocked down by Reactome/PADOG ^1^.

Pathway	COLO357–siSOX9		PANC-1–siSOX9	
	Av. FC ^2^	Number of Genes	Av. FC	Number of Genes
Up-regulated pathways				
Regulation of TP53 activity through acetylation (R-HSA-6804758)	5.09	6	19.63	11
Erythropoietin activates phospholipase C gamma (PLCG) (R-HSA-9027277)	28.57	1	12.49	4
Erythropoietin activates STAT5 (R-HSA-9027283)	28.57	1	5.31	4
Organic anion transporters (R-HSA-428643)	15.01	1	14.81	4
Defective SLC17A5 causes Salla disease (SD) and ISSD (R-HSA-5619035)	15.01	1	46.81	1
Down-regulated pathways				
MAPK6/MAPK4 signaling (R-HSA-5687128)	−17.95	15	−8.011	35
Opposite regulated pathways				
Erythropoietin activates phosphoinositide-3-kinase (PI3K) (R-HSA-9027276)	4.75 (up)	3	−0.12 (down)	7
Tandem pore domain potassium channels (R-HSA-1296346)	−16.0 (down)	1	21.05 (up)	1
Tandem of pore domain in a weak inwardly rectifying K+ channels (TWIK) (R-HSA-1299308)	−16.0 (down)	1	21.05 (up)	1
Phase 4—resting membrane potential (R-HSA-5576886)	−16.66 (down)	3	12.03 (up)	2
TP53 regulates transcription of cell death genes (R-HSA-5633008)	−0.35 (down)	8	4.59 (up)	24
Alternative complement activation (R-HSA-173736)	−39.74 (down)	1	N.i. ^3^	N.i.
Activation of C3 and C5 (R-HSA-174577)	−39.74 (down)	1	N.i.	N.i.

^1^ Processes with FDR (false discovery rate) ≤ 0.001 for both cell lines only are presented. ^2^ Average fold change (log) by Reactome-PADOG. ^3^ Not informative.

#### 2.2.4. GSEA Expression Analysis

The GSEA analysis involving MSigDB Hallmarks gene sets [27] using the GSEA 4.3.2 program [28] (Appendix A) was also performed. Table 6 and Appendix A summarize the Hallmark gene sets with which overlaps were identified (Hallmarks with the most conservative Family-wise error rate FWER *p*-val < 0.05 are shown). According to the results of the analysis, increased expression of genes regulated by MYC oncogene and genes suppressed by UV irradiation was found in COLO357 cells with suppressed SOX9 expression relative to control cells. At the same time, COLO357–siSOX9 cells show decreased expression of late estrogen response genes. In PANC-1 cells, *SOX9* knockdown acts reciprocally to KRAS oncogene activation and suppresses expression of E2F transcription factors targets. Intersections with oncogene MYC targets, UV response genes, and late estrogen response genes for PANC-1–siSOX9 cells are statistically insignificant (Appendix A).

Intersections of gene sets that changed expression upon *SOX9* knockdown in PAAD cells with a number of other important Hallmark gene sets, including Apical junction set, p53 pathway set, epithelial–mesenchymal transition set, inflammatory response, and differentially expressed genes in PAAD [29] were found, but correlations for these intersections did not exceed thresholds of significance.

**Table 6 ijms-26-02652-t006:** MSigDB Hallmark gene sets expression change in PAAD cells with *SOX9* knocked down ^1^.

Hallmark Name	SIZE	ES	NES	NOM *p*-Val	FDR q-Val	FWER *p*-Val
COLO357–siSOX9 up-regulated Hallmark gene sets						
MYC_TARGETS_V1 ^2^	36	0.39	2.28	0	0.0031	0.002
MYC_TARGETS_V2 ^2^	36	0.43	2.22	0	0.0022	0.003
UV_RESPONSE_DN ^2^	25	0.38	1.91	0.0071	0.018	0.03
COLO357–siSOX9 down-regulated Hallmark gene sets						
ESTROGEN_RESPONSE_LATE ^2^	57	−0.42	−2.03	0	0.015	0.019
PANC-1–siSOX9 up-regulated Hallmark gene sets						
KRAS_SIGNALING_DN ^2^	30	0.40	1.78	0.071	0.016	0.035
PANC-1–siSOX9 down-regulated Hallmark gene sets						
E2F_TARGETS ^2^	118	−0.42	−2.06	0	0.042	0.04
(KRAS_SIGNALING_UP) ^2^	54	(−0.43)	(−1.82)	0	(0.030)	(0.053)

^1^ Size—number of expressing genes of certain set; ES—enrichment score, NES—normalized enrichment score; NOM *p*-val (nominal *p*)—the statistical significance of ES not adjusted for gene set size; FDR q-val—false discovery rate; FWER *p*-val—familywise-error rate. Hallmark sets with FWER *p*-val < 0.05 are presented with exception for KRAS_SIGNALING_UP. Full gene sets names are presented in Appendix A. ^2^ See Appendix A.

#### 2.2.5. Oncogenes, Tumor Suppressors, EMT Genes

We compared the results of *SOX9* knockdown in PAAD cells with oncogene signature gene sets (MSigDB-C6 group), which are sets of genes often dysregulated in cancer.

The analysis of COLO357 cells with *SOX9* knockdown reveals increased expression of genes activated in breast cancer with MYC gene overexpression, as well as decreased expression of genes under-expressed in colon cancer cells, with artificial overexpression of the *LEF1* gene included in epithelial–mesenchymal transition (EMT), and decreased expression of genes up-regulated in CD34+ cells following the knockdown of the *RPS14* gene, a ribosomal protein with distinct onco-promoting effects [30,31] and a negative regulator of MYC [32].

For PANC-1–siSOX9 cells relative to PANC-1–siNEG cells, increased expression of genes activated by *PTEN* oncogene knockdown and genes suppressed by treatment of epithelial cells with TGFB1 factor was revealed. At the same time, the expression of genes activated in fibroblasts by knockdown of oncogene *EZH2* and genes suppressed in MCF10A cells (breast cancer) by knockdown of *RBBP8/CtIP* gene, the product of which is a cofactor of BRCA1 in repair of DNA double-strand breaks [33], was decreased in PANC-1 cells with *SOX9* knockdown (Table 7, Appendix A).

The differentially expressed upon *SOX9* knockdown genes were also matched with sets of genes involved in EMT from the dbEMT2 database [34]. The *SOX9* gene is not included in the dbEMT2 lists; however, its cognate genes *SOX2* and *SOX4* are listed among the oncogenes. As shown in Figure 5a,b and Appendix A, both over- and under-expressed genes in each group are observed in cells of both cell lines upon SOX9 knockdown relative to corresponding control cells. Among the common genes for cells of both lines are the oncogenes *HSPB1*, *IRS2* (up-regulated), and *FOXQ1* (down-regulated), as well as the suppressor genes *AHR*, *DKK1* (up-regulated), and *NUAK1*, *CTNNBIP1*, *HIPK2*, and *H2AX* (down-regulated).

Taking into account the fact that SOX proteins may interact directly or indirectly (see Introduction), the effect of *SOX9* knockdown on the expression of other members of SOX family was additionally estimated. No significant change in the expression of other *SOX* genes in COLO357 cells with *SOX9* knockdown, compared to control cells, was observed, except for *SOX9* itself. In PANC-1–siSOX9, the suppression of *SOX12*, *SOX21*, and *SOX21-AS* expression relative to control cells was identified (Appendix A). Separately, the pancreatic cancer EMT signature genes were investigated [35]. Of the 22 genes in this set, informative data were obtained for 5 genes (Figure 5c). A significant change (decrease) in the expression in both cell lines was found only for the *KRT7* gene. Moreover, in Colo-357–siSOX9 cells, a decrease in the expression of *CLDN4* and *MUC* genes was observed, and in PANC1–siSOX9 cells, a decrease in the expression of *EPCAM* gene and an increase in the expression of *SPARC* gene were identified relative to corresponding control cells. Therefore, the obtained data do not permit the establishment of a conclusion regarding a particular alteration in the expression of a specific group of epithelial–mesenchymal transition genes (oncogenes, suppressors, or dual-action genes) in response to *SOX9* suppression in the examined PAAD cells.

#### 2.2.6. P53 Gene-Related Alterations

As discussed in the Introduction, COLO357 cells possess the wild-type *p53* gene, whereas in PANC-1 cells, codon 273 of the p53 gene is mutated (CGT- > CAT), resulting in amino acid substitution of p.Arg273His [25,26]. *SOX9* knockdown in the cells lunes under examination, primarily PANC-1, affected multiple processes involving p53 regulation (Table 8). However, according to Deseq2 results, neither in Colo-357–siSOX9 nor in PANC-1–siSOX9 did the change in p53 expression exceed the thresholds of significance relative to corresponding cells transfected by siNEG RNA.

#### 2.2.7. PDAssigner

A number of alterations in genes from the PDAssigner gene set [24] were observed both in COLO357 and PANC-1 cell lines under *SOX9* knockdown. As mentioned above, the authors of PDAssigner assigned the cells of both cell lines to the QM-PDA subtype, although PANC-1 cells exhibited closer proximity to the epithelial phenotype than COLO357 cells in terms of sensitivity to cytostatics. However, the comparison of PANC-1–siNEG and COLO357–siNEG cells both transfected with neutral siRNA reveals the greater levels of expression of genes associated with the epithelial phenotype in COLO357 cells compared to PANC-1 cells (Figure 6a).

Following *SOX9* knockdown, a decline in the expression of multiple classical PDA-specific genes was observed in COLO357 cells. (Figure 6b; Appendix A). At the same time, the expression of some QM-PDA subtype genes was markedly decreased in PANC-1–siSOX9 cells. However, no increase in the expression of genes characteristic of classical PDA was observed in PANC-1–siSOX9 cells, nor was there an increase in the expression of QM genes in COLO357–siSOX9 cells relative to control cells. Therefore, it can be concluded that SOX9 suppression does not shift the characteristics of COLO357 cells toward mesenchymal and PANC-1 cells toward classical PDA.

Furthermore, a significant decrease in *FOXQ1* oncogene expression was observed in both COLO357–siSOX9 and PANC-1–siSOX9 cell lines compared to the corresponding cells transfected with neutral siRNA.

#### 2.2.8. Transcription Factors

Given the established role of SOX9 as a master regulator of transcription for numerous genes, we investigated the alterations in transcription factor (TF) expression levels in PAAD cells with SOX9 knockdown. For this purpose, we used The Human Transcription Factors database ver. 1.01, which contains data on 1639 transcription factors [36]. In the case of COLO357–siSOX9 cells, alterations in transcription levels of 26 TFs were revealed, with increased levels found for 9 TFs and decreased levels for 17 TFs. In PANC-1–siSOX9 cells, the expression levels of 209 TFs were altered, of which 138 TFs had increased transcription levels and 71 factors were down-regulated (Appendix A). The transcription factors with altered expression in cells of both lines upon SOX9 knockdown are shown in Figure 7a. Some of the important processes in which the identified transcription factors are involved are summarized in Table 9.

#### 2.2.9. Genes Bound to SOX9 by ChIP

Shih at al. discovered 3730 genes binding SOX9 in chromatin immunoprecipitation [37]. We examined alterations of the expression of these genes in COLO357 and PANC-1 cells upon SOX9 knockdown. The expression of 130 genes was altered in COLO357–siSOX9 cells, and the expression of 418 genes was altered in PANC-1–siSOX9 cells relative to the corresponding cells transfected with neutral siRNA (siNEG RNAs) (Appendix A). The expression of 51 genes changed was altered in both investigated cell lines upon *SOX9* knockdown, of which the expression of 11 genes increased and 35 genes decreased. The expression of five genes exhibited divergent alterations (Figure 7b, Section 2.2.8).

Using Reactome database, we conducted the examination of the processes involving genes that demonstrated similar alterations in both COLO357 and PANC-1 cells following *SOX9* knockdown. Among the processes that were activated in both cell lines, processes involving *NGFR* and the genes *TAF13* and *SGK1*, which function as regulators of p53 activity and transcription, were particularly prominent. Conversely, signal transduction and cell cycle regulation processes involving *E2F7*, *CDC6*, *CKS1B*, and other genes were suppressed (Appendix A).

### 2.3. Analysis of Pancreatic Cancer Crucial and Prognostic Gene Sets

#### 2.3.1. Differentially Expressed Genes in Pancreatic Cancer

Previously, a number of investigators have identified genes that are differentially expressed in human PAAD (approximately 800 overexpressed genes and approximately 400 under-expressed genes) compared to the normal pancreas [29,38,39,40,41,42,43]. In this study, we investigated the effect of *SOX9* knockdown on the expression of these genes. The investigation reveals that 50 genes overexpressed in PAAD exhibit altered expression levels in response to *SOX9* knockdown for both studied cell lines. Additionally, the expression levels of 26 under-expressed genes in PAAD were influenced by *SOX9* knockdown in the examined cell lines (Appendix A). It should be noted that among both groups of differently expressed in PAAD genes, there are both activated and suppressed genes under *SOX9* knockdown (Figure 8). For example, the expression of *SELENOW* (selenoprotein W) gene is up-regulated in PAAD according to Grutzmann et al. [29] and down-regulated according to Ryu et al. [42].

Additionally, we will reference the gene sets with increased (363 genes) and decreased (204 genes) expression in PAAD identified by Grutzmann et al. [29]. Separate analysis of genes from these sets that changed expression in cells during *SOX9* knockdown using GSEA analysis did not reveal statistically significant patterns (see Section 2.2.4).

#### 2.3.2. Prognostic Gene Signatures

A number of gene expression signatures that are associated with the patient prognosis, sensitivity to anticancer drugs, or individual features of pancreatic cancer have been published. We have analyzed if the expression signatures utilized in these studies are altered in COLO357 and PANC-1 cells upon SOX9 knockdown (Table 10, Figure 9a). Only several genes of the examined signatures changed expression upon SOX9 knockdown.

**Table 10 ijms-26-02652-t010:** Selected expression gene signatures for pancreatic cancer.

Pancreatic Cancer Gene Signature	Reference	Number of Genes in a Signature	Genes Affected Under SOX9 Knock-Down	
			COLO357–siSOX9	PANC-1–siSOX9
Gene signature of metastasis	[44]	17	No genes *	*COL1A1*, *EIF4E2*
Gene signature for metastasis and survival	[45]	6	No genes *	No genes *
Patients outcome gene expression signature	[46]	36	No genes *	*ADM*, *ITGBL1*, *CDK2AP1*, *ARRB1*, *SEMA3A*, *PHLDA1*, *BLM*, *KIF14*, *PXN*
Survival gene expression signature	[47]	19	No genes *	*AVPI1*, *CLIP4*, *RGS20*
Genetic predictors for survival under chemotherapy	[48]	23	***CDRL2***, *RFNLA*	* **CDRL2** *
Gene expression signature for prognosis	[49]	5	* **E2F7** *	* **E2F7** *
KRAS-associated metabolic genes for prognosis	[50]	6	* **GPX3** *	* **GPX3** *
An EMT-related gene signature for predicting response to adjuvant chemotherapy	[51]	8	*ITGB6*	*DLX2*
Prognostic gene signature	[52]	18	*CD9*, ***UBASH3B***, *FYN*	*GNA15*, ***UBASH3B***, *SERPINE1*
Metabolism-related genes as targets for immunotherapy	[53]	24	*SLC1A3*	*IGFBP6*, *HK2*, *SLC2A1*, *ALDH1A3*
MYC-based signature for prognosis and chemoresistance	[54]	7	*ITGB6*	No genes *

* Genes were filtered out by Deseq2 program or FC < 1.5. Common genes are written in bold.

Separately, we examined the alterations in the expression of 171 genes within an integrative model of patient survival based on the expression levels of marker genes [55]. This model encompasses 74 up-regulated genes and 97 down-regulated genes in PAAD. In COLO357 and PANC1 cells with suppressed *SOX9* expression, among the genes from both groups, the expression of some genes increased and other genes decreased (Figure 9b). Consequently, no specific patterns of changes in the expression of the aforementioned signatures in COLO357 and PANC-1 cells upon *SOX9* knockdown were revealed.

We also compared the RNA-seq results of the cells with *SOX9* knockdown to pancreatic cancer and metastasis marker sets proposed by Missiaglia et al. [38]. PANC-1 and COLO357 cells with *SOX9* knockdown exhibit a decrease in the expression of a group of genes whose up-regulation is characteristic of primary tumors (cluster A) and genes whose expression is up-regulated in PAAD cell lines isolated from metastases to lymph nodes (cluster C)—in particular, the *MEGF6* gene (Figure 9c,d). This finding suggests a potential decrease in tumorigenic potential of PAAD cells upon SOX9 knockdown. In addition, an increase in the expression of *SGK1* (cluster F, transport/signal transduction) and *IRS1* (cluster I, cell communication/signal transduction) was observed in both cell lines upon *SOX9* knockdown. This may indicate the activation of intercellular interaction through molecular signals.

**Figure 9 ijms-26-02652-f009:**
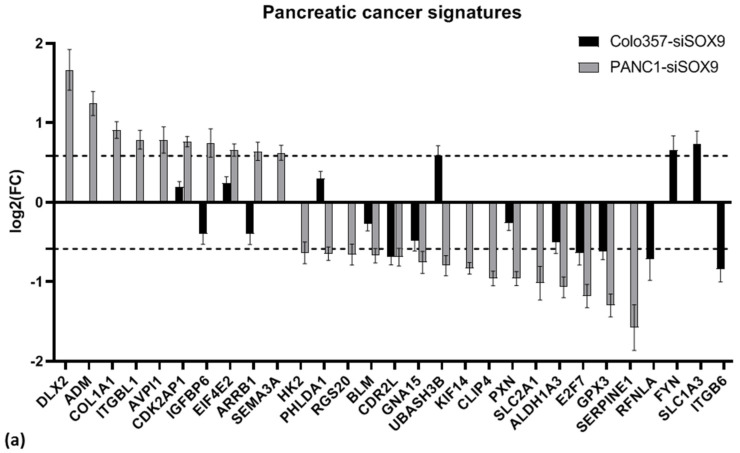
Alterations of the expression of genes from pancreatic cancer gene expression signatures (references are in Section 2.3.2) in COLO357 and PANC-1 cells under SOX9 knockdown relative to corresponding control cells; (**a**) genes from selected expression signatures (Table 10), (**b**) integrative PAAD survival model genes [55]; for (**a**,**b**), genes with fold change FC > 1.5 (dashed line) for any one line are shown, (**c**,**d**) tumorigenesis and metastasis marker genes [38] in COLO357 (**c**) and PANC-1 (**d**) cells with SOX9 knockdown relative to corresponding control cells. Clusters A–D, F, G and I on horizontal axis are indicated by Missiaglia et al. [38]. For (**c**,**d**) genes with FC > 1.5 in both cell lines (dashed lines) are presented only.

#### 2.3.3. Drug Resistance Genes

We examined the expression changes of 465 genes whose expression was reported as associated with resistance or sensitivity to gemcitabine according to sources in the literature [47,49,50,56,57,58,59,60] in COLO357 and PANC-1 cells with *SOX9* knockdown (Figure 10). In total, 17 genes associated with gemcitabine resistance and 6 genes associated with sensitivity to this drug were down-regulated in COLO357–siSOX9 cells, while no up-regulation was detected among genes from either group. In PANC-1–siSOX9 cells, among genes associated with resistance to gemcitabine, we found increased expression of 19 genes and decreased expression of 14 genes; among genes associated with sensitivity to this drug, we found increased expression of 15 genes and decreased expression of 23 genes (Appendix A). These differences showed a level of statistical significance of *p* > 0.1 in Fisher’s and chi-square tests.

A similar pattern, namely lack of significant difference between quantities of sensitivity and resistance genes, emerged for genes associated with the response of PAAD organoids to paclitaxel, a topoisomerase I inhibitor SN38, 5-fluorouracil, and oxaliplatin [56] (Figure 10). Meanwhile, COLO357 and PANC-1 cells with SOX9 knockdown show decreased expression of *HELZ2* and *TMEM9B* (paclitaxel resistance); *GIT1*, *RNF144A*, and *PGD* (paclitaxel sensitivity); *ZBTB46* and *TMEM9B* (SN38 resistance); *NF2* and *PPP2R1B* (SN38 sensitivity); *ZBTB46* (5-FU resistance); *HSPA* (oxaliplatin resistance); and *SFXN5* and *SECISBP2* (oxaliplatin sensitivity) relative to corresponding control cells.

SOX9 suppression in COLO357 cells led to increased expression of the single *HIVEP2* gene associated with resistance of PAAD organoids to oxaliplatin, while other genes tested were repressed or their expression did not demonstrate significant difference from corresponding control cells. In PANC-1–siSOX9 cells, a part of genes associated with positive or negative response to any of five drugs, including gemcitabine, had increased expression, and a part of genes had decreased expression relative to control cells (Appendix A). We did not find significant difference between these groups of genes in case of PANC-1 cells.

Separately, we analyzed expression of ATP-binding cassette (ABC) superfamily proteins. Only *ABCB10* gene expression was suppressed in both COLO357 and PANC-1 cells transfected with SOX9-specific siRNA relative to corresponding cells transfected with neutral siRNA (Figure 10). In Colo357–siSOX9 cells, *ABCB1* gene was up-regulated; in PANC-1–siSOX9 cells, *ABCA1* and *ABCA2* genes expression was elevated compared to corresponding cells transfected with neutral siRNA (Appendix A). Thus, SOX9 suppression had a mixed effect on the expression of ABC genes.

## 3. Discussion

In this study, we investigated the biological functions and prognostic significance of the SOX9 gene, one of the members of the SOX factor family. SOX9 transcription factor is involved in many signaling pathways important in carcinogenesis, including the Fgf, Notch, Wnt/beta-catenin, NF-kB, and Hippo pathways [3,19,61,62,63]. In particular, the EGF–SOX9–TSPAN8 signaling cascade has been shown to be involved in the control of invasion of ductal adenocarcinoma of the pancreas [64].

*SOX9* gene expression is known to be altered during the development of pancreatic tumors. *SOX9* expression level has been shown to correlate with drug resistance and malignant potential of tumor cells [17,18]. SOX9 activation in acinar cells in chronic pancreatic inflammatory processes leads to acinar ductal metaplasia (ADM), which is an essential event in the early stages of pancreatic carcinogenesis [65]. In precancerous states, increased Sox9 activity ensures the survival and proliferation of cells with mutant Kras oncogene in mice [66]. In pancreatic ductal cells and their precursors, the *KRAS* mutant oncogene constitutively activates the expression and translocation of SOX9 to the nucleus involving the TAK1/IκBα/NF-κB signaling pathway [67].

We have previously shown that significant changes occur in PAAD cells upon *SOX9* knockdown that differ in different cell types [16]. In particular, COLO357–siSOX9 cells showed a decrease in the expression of epithelial markers CDH1, KRT7, KRT18, KRT19, ZO-1, and others relative to control cells, while in PANC-1–siSOX9 cells, no pronounced decrease in the expression of the epithelial markers was detected. Changes in the expression of a number of mesenchymal markers were equivocal. Both in COLO357 and PANC-1 cells with SOX9 knockdown, there was a decreased expression of developmental markers—transcription factors GATA4, GATA6, and FOXA2. Also, the expression of some cell cycle regulators (CCND3, CDKN1A) increased in cells of the two mentioned lines. Increased expression of p21Waf/Cip, p27Kip (CDKN1B), p53, and PTEN with decreased expression of cyclin B1 (CCN1B), survivin (BIRC5), and BMI1 was found in PANC-1–siSOX9 cells relative to control cells. In COLO357, PANC-1, and some other cell lines with suppressed SOX9, significant inhibition of proliferation, as well as increased functional activity of caspases 3/7, 8, and 9, were observed. For PANC-1 cells with *SOX9* knockdown, an increase in the number of cells in a state of apoptosis and a decrease in migration activity upon xenotransplantation into *Danio rerio* were shown.

It should be emphasized that the effect of SOX9 on proliferation differs in different pancreatic cancer cells and in different tumors. For example, the work cited above found only a slight decrease in proliferation of MiaPaCa-2 cells upon *SOX9* knockdown. For cervical cancer cells, a suppressor role of SOX9 has been shown to suppress tumor growth and tumorigenicity when SOX9 is overexpressed [12], but in esophageal squamous cell cancer, SOX9 overexpression leads to increased proliferation and tumorigenicity [68].

To investigate the function and importance of SOX9 for tumor prognosis and therapy further, we performed a full transcriptome analysis of PANC-1 and COLO357 pancreatic cancer cells with SOX9 activity suppressed by RNA interference.

### 3.1. Primary Analysis of Transcriptomes

Primary cell transcriptome analysis shows that PANC-1 cells originating from primary PAAD responded much more vividly to SOX9 suppression than COLO357. The total number of genes with altered expression in PANC-1 cells with SOX9 knockdown (relative to corresponding cells transfected with neutral siNEG siRNA) was approximately 2.5-fold greater than in COLO357 with *SOX9* knockdown. Interestingly, most of the genes that changed expression more than 1.5-fold in PANC-1–siSOX9 cells, changed expression in the same direction in COLO357 cells with *SOX9* knockdown, although they often did not reach the FC > 1.5 threshold level relative to the corresponding control cells. Taken together with the evidence of higher SOX9 expression levels in PANC-1 cells than in COLO357 cells [16], this may imply a greater activity of SOX9-regulated processes and a greater dependence of PANC-1 cells in general on the SOX9 factor, compared to COLO357 cells. This possibly indicates a loss of importance of SOX9-regulated processes or a loss of the ability of these processes to respond to SOX9 master regulation during metastasis. The relatively small number of identical genes with altered expression in PANC-1 and COLO357 cells with *SOX9* knockdown also indicates significant differences between metastatic cells and primary PAAD cells. One other reason for the significant difference in cellular response to *SOX9* knockdown may be that the p53 gene is wild-type in COLO357 cells, while in PANC-1 cells the p53 gene has a p.Arg273His mutation. Additionally, the differences may have accumulated in the genomes of the cells during cultivation.

### 3.2. Pathway and Hallmarks Analysis: Proliferation and Cell Cycle

Examination of biochemical processes in PAAD cells using expression databases reveal both general and cell line-specific processes that changed activity under SOX9 knockdown.

Gene Ontology annotation analysis in the present work shows that genes that responded to intracellular stimuli and signal transduction were activated in COLO357–siSOX9 cells relative to control COLO357–siNEG cells. In PANC-1–siSOX9 cells, genes regulating developmental processes, transport, transcription, biosynthesis, and differentiation were most active (Figure 2) relative to PANC-1–siNEG control cells.

SOX9 knockdown in COLO357 and PANC-1 cells activated some genes of the SOS signaling pathway (R-HSA-112412, Table 2). As indicated above, cells from both cell lines carry a mutant KRAS oncogene. The involvement of the SOS-mediated pathway is consistent with our data on the changes in the expression of genes activated by KRAS oncogene in PANC-1–siSOX9 cells (see Table 6), as well as with the previously proposed model of positive RAS/SOS feedback [69] and also with the data on the activation of NRAS and wild-type HRAS through allosteric mutKRAS/SOS1 interaction, which gives a growth-enhancing effect on the cells with mutated KRAS [70].

In both COLO357 and PANC-1, *SOX9* knockdown activates genes in the “POU5F1 (OCT4), SOX2, NANOG repress genes related to differentiation” pathway. The POU5F1 (OCT4), SOX2, and NANOG transcription factors cooperatively regulate the expression of a large group of transcription factor genes responsible for embryonic stem cell pluripotency [71]. This may indicate the involvement of SOX9 in this regulation and is consistent with a guiding role of SOX9 in embryonic organogenesis [3,19].

The pathway “Replacement of protamines by nucleosomes in the male pronucleus” is relevant during oocyte fertilization, where protamines occupying sperm genomic DNA are replaced by nucleosome histones provided by the oocyte, leading to chromatin decondensation and promoting its transcription [72,73]. This may reflect chromatin decondensation in PAAD cells with SOX9 knockdown for transcription of some genes inactive in the original PAAD cells, or a general tendency for tumors to exhibit embryonic features and occasional signs of deregulation. Also upon SOX9 knockdown, genes involved in telomere end packaging, regulation of ribosomal RNA synthesis (RNA polymerase I promoter opening) and DNA methylation were activated in PAAD cells. This may have also contributed to changes in the expression levels of many genes and a decrease in the proliferative activity of cells.

Using Reactome PADOG analysis, we show that positive regulation of p53 (in particular, by acetylation) was activated in COLO357 and PANC-1 cells with suppressed SOX9 (Table 5). This may enhance p53 binding to target gene promoters and induction of apoptosis [74]. Increased expression of genes taking part in processes involving erythropoietin—activation of phospholipase C gamma and STAT5—has also been observed; these processes are important for activation of calcium signaling [75] and gene expression [76], respectively.

Among the processes suppressed in SOX9 knockdown in PANC-1 cells, the most notable are those related to DNA replication, regulation of various stages of the cell cycle, and apoptosis (Figure 2, Table 2). Interestingly, *SOX9* knockdown simultaneously activated developmental and differentiation-related processes in PANC-1 cells, which may indicate a decrease in the tumorigenic potential of the cells. Meanwhile, Gene Ontology analysis of the annotations for COLO357–siSOX9 cells did not reveal any noticeable change in the expression of the corresponding genes. At the same time, down-regulated expression of cell adhesion related genes were found in COLO357–siSOX9 cells. This may contribute to the decrease in cell migration activity upon *SOX9* suppression noted in our previous work [16].

Both COLO357–siSOX9 and PANC-1–siSOX9 have reduced expression of genes that determine the G1 phase of mitosis and G1/S phase transition relative to corresponding control cells (Table 2). The reduced activity of genes involved in cell cycle regulation (e.g., G1 and G1/S transition phases) upon SOX9 knockdown is in good agreement with the previously obtained data on the significant reduction in COLO357 and PANC-1 proliferation upon *SOX9* suppression [16]. At the same time, genes determining almost all cell cycle stages—R-HSA-68875:mitotic prophase, R-HSA-69275: G2/M transition, and so on—were markedly suppressed in PANC-1–siSOX9 cells (Appendix A). This effect was less pronounced in COLO357–siSOX9 cells.

In addition to suppression of proliferation, COLO357 and PANC-1 cells with *SOX9* knockdown showed reduced expression of genes involved in apoptosis, including SMAC- and XIAP-mediated processes, as well as activation of caspases by cutting with apoptosome (in this case, decreased expression of CASP7 was found). The difference in expression of other caspases between PAAD cells with *SOX9* knockdown and control cells did not exceed threshold levels. This observation contrasts with our previous findings of functional activation of caspases 3/7, 8, and 9 in BxPC-3, COLO357, and PANC-1 cells with *SOX9* knockdown [16], as well as the increased number of Caspase 3+ cells in PANC-1 culture after *SOX9* knockdown compared to control cells found by Aldaz et al. [77].

At the same time, in the present work, activation of genes related to cell senescence (*H2BC12*, *IL6*, *MAP2K6*) was found in COLO357–siSOX9 cells, and increased expression of FOXO-mediated transcription of cell death genes (*PINK1*, *CREBBP*, *BCL2L11*, *CITED2*) was found in PANC-1–siSOX9 cells (Appendix A). Thus, SOX9 knockdown, in addition to reducing cell proliferation, can probably promote tumor cells to their death without involving caspases. This phenomenon needs to be investigated in more detail.

MAPK6/MAPK4 signaling genes were suppressed in COLO357 and PANC-1 cells with SOX9 knockdown (Table 5). MAPK4 and MAPK6 belong to atypical mitogen-activated protein kinases and are able to form functional heterodimers. It has been shown that MAPK6 (ERK3) depletion markedly decreased oncogenic growth of mutant KRAS-mediated anchorage-independent growth of NSCLC cells [78], but no such data are available for PAAD. Since COLO357 and PANC-1 cells contain mutant KRAS oncogene, our data are in favor of such an effect in PAAD.

As is was mentioned in the Introduction, COLO357 and PANC-1 cells have different origins. Therefore, we investigated the most striking differences in the transcriptomes of these cells after *SOX9* knockdown. COLO357–siSOX9 cells showed activation of response genes to intracellular stimuli and down-regulated expression of cell adhesion regulation genes (Figure 2). At the same time, in PANC-1 cells, *SOX9* suppression revealed more involvement of developmental, differentiation, and transport processes for activated genes, and many proliferation processes and a greater number of cell cycle stages for genes with reduced expression.

In addition, a large group of processes specific to both COLO357 and PANC-1 with suppressed *SOX9* were identified (Figure 3; Appendix A).

We also identified processes that are activated in COLO357–siSOX9 cells and suppressed in PANC-1–siSOX9 cells. In particular, these are the processes related to cell aging and activation of protein kinases N by RHO GTPases (Table 3).

Analyzing effects of *SOX9* knockdown on immunity-related genes in pancreatic cancer cells, we received somewhat contradictory results their estimates depending on statistical tests used.

Using Reactome representation analysis, we studied inflammation and immunity pathways and found as many activated as repressed pathways in cells with SOX9 knockdown. Thus, in COLO357 cells with SOX9 knockdown, such pathways as signaling by interleukin 4, 12, 13, and 35 are up-regulated while chemokine binding by receptors are down-regulated relative to control cells (Appendix A). Moreover, pathways R-HSA-6783783 interleukin-10 signaling and R-HSA-449147 signaling by interleukins both contain genes expression that changed in opposite directions upon SOX9 knockdown in COLO357 cells (Table 4). In PANC-1 cells, *SOX9* knockdown resulted in up-regulation of insulin receptor recycling pathway but in down-regulation of interleukin-33 signaling (Appendix A).

Reactome PADOG analysis reveals interleukin 12 signaling among top 10 up-regulated pathways in COLO357–siSOX9 cells when calculated by number of involved genes, while TNF receptor superfamily (TNFSF) members mediating non-canonical NF-kB pathway were the most down-regulated pathways in these cells on the basis of expression fold change (Appendix A). In PANC-1–siSOX9, innate immune system pathway was the most suppressed pathway when counting by gene names (375 genes); this pathway is significant but far from the top down-regulated ones if we consider expression fold change (Appendix A).

In GSEA analysis, down-regulation of genes up-regulated by IL6 via STAT3 (HALLMARK_IL6_JAK_STAT3_SIGNALING) in COLO357 and PANC-1 cells with SOX9 knockdown can be recognized as significant using FDR q-val criterion but not FWER-pVal accepted by us in this case (Appendix A). The same goes for genes regulated by NF-kB in response to TNF (HALLMARK_TNFA_SIGNALING_VIA_NFKB), which were activated in COLO357–siSOX9 cells compared to control cells.

Thus, we may conclude that SOX9 knockdown affects expression of genes related to immunity in pancreatic cancer cells and that the biological sense of such influence should be additionally studied.

### 3.3. Oncogenes and Tumor Suppressor Genes

By analyzing the annotations of differentially expressed genes using modern databases, we show that the functions of SOX9 are closely related to those of a number of oncogenes. Thus, in both cell lines with *SOX9* knockdown, the expression of *HSPB1* and *IRS2* oncogenes was up-regulated, and the expression of *FOXQ1* oncogenes was reduced (Figure 5). HSPB1 is a protein from the small heat shock protein family with chaperone functions; its increased expression in some cancers promotes tumor progression and metastasis, and correlates with tumor chemoresistance and poor prognosis, although its role in the case of PAAD is controversial [79,80]. IRS2 is an intermediate in the signal transduction chain from insulin and other cytokines via tyrosine kinase receptors to other effectors. Increased IRS2 expression in colorectal cancer promotes tumor progression and invasion by activating the PI3 kinase pathway [81]. FOXQ1 promotes pancreatic cancer cell proliferation, tumor stemness, invasion, and metastasis through regulation of LDHA-mediated aerobic glycolysis [82]. Thus, SOX9 suppression in the case of these three oncogenes has a contradictory effect, enhancing some drivers of tumor growth and suppressing others.

Examination of the expression of oncogenes, suppressor genes, and genes with dual functions involved in epithelial–mesenchymal transition and included in dbEMT database show that in COLO357 and PANC-1 cells with *SOX9* knockdown, among the genes of each group there are both activated and repressed genes relative to the corresponding control cells in approximately proportional amounts (Figure 5).

Using Reactome, COLO357 and PANC-1 cells with SOX9 knockdown were shown to have activated the SOS-mediated signaling pathway for activation of RAS (R-HSA-112412, Table 5). Of note, both cell lines studied contain an activated mutant KRAS (p.Gly12Asp), one of the most frequent and earliest mutations in RAS [83]. The relationship between KRAS and SOX9 has been reported previously [84]. Both activated mutant KRAS and SOX9 overexpression promote aberrant cell proliferation, with KRAS constitutively activating the expression and translocation to the nucleus of SOX9 involving the TAK1/IκBα/NF-κB signaling pathway [67]. Our data suggest that suppression of SOX9 in PANC-1 cells activates genes repressed by mutant KRAS and conversely represses genes activated by mutant KRAS (see Panther pathway, Appendix A, GSEA Section 2.2.4, Table 6, Appendix A). This indicates the presence of both positive and negative regulation between KRAS and SOX9. At the same time, the expression of KRAS itself changes insignificantly upon *SOX9* knockdown in the cells under study.

The set of genes activated in *SOX9* knockdown COLO357 cells largely overlaps with those activated by the *MYC* oncogene, a major regulator of the cell cycle (Table 6 and Table 7, Appendix A). C-Myc contributes to tumor angiogenesis, metastasis, and chemoresistance; C-Myc overexpression is associated with chemoresistance, intra-tumor angiogenesis, epithelial–mesenchymal transition (EMT), and metastasis in pancreatic cancer [85]. One might expect that SOX9 suppression might produce similar effects independently of MYC, but our results do not support this. Also, it was shown that c-Myc can increase the transcription of WNT-related genes such as lymphoid enhancer-binding factor 1 (LEF1) and augment WNT/β-catenin signaling pathway in cancer cells [85]. *LEF1* activation is known to induce EMT in colorectal carcinoma cells [27]. However, our data show that *SOX9* knockdown in COLO357 also suppressed the expression of genes repressed by overexpression of the *LEF1* oncogene (Table 7). Perhaps LEF1 is one of the key indicators of the difference between the effects of MYC activation and *SOX9* knockdown and an example of the ambiguous effect of SOX9 on EMT in PAAD. However, *MYC* and *LEF1* gene expression are not significantly altered by *SOX9* knockdown in cells from both cell lines studied.

COLO357–siSOX9 has down-regulated expression of genes activated by suppression of the RPS14 oncogene. RPS14 (ribosomal protein S14) is thought to promote the progression of some tumors, particularly liver cancer via the Akt signaling pathway [86] and glioma via the p53 signaling pathway [31]. This context may indicate an onco-promoting role of SOX9 suppression (respectively, a suppressor role of SOX9), which calls its use as a therapeutic target questionable in PAAD.

In PANC-1–siSOX9 cells, the set of activated genes overlaps significantly with genes activated upon knockdown of the *PTEN* oncogene [87]. Thus, SOX9 and PTEN act in a similar manner, indicating a possible link between SOX9 and the JNK signaling pathway. At the same time, the expression of *PTEN* itself was up-regulated in PANC-1 cells with SOX9 knockdown relative to control cells (log2FC = 0.69) while in COLO357–siSOX9 *PTEN* expression change did not exceed threshold levels relative to that in COLO357–siNEG cells.

In PANC-1 cells with suppressed SOX9 expression, expression of genes activated by knockdown of the *EZH2* oncogene was suppressed. The EZH2 protein is a member of the polycomb-repressive complex 2 (PRC2). Polycomb group (PcG) proteins form chromatin-modifying complexes that are essential for embryonic development and stem cell renewal, and are commonly deregulated in cancer [88]. Our data suggest that *SOX9* knockdown also activates several processes related to differentiation and development in PANC-1 cells (Figure 2 and Figure 3).

SOX9 acts similarly to the CtIP protein (*RBBP8*) because the sets of genes repressed upon knockdown of these two genes in PANC-1 cells overlap (Table 7). The *RBBP8* gene is known to be amplified in 16% of PAADs and its overexpression correlates with a poor prognosis [89]. The CtIP protein is part of a repressor complex that binds to the promoter of the ANG1 gene [90] and also represses p21 protein expression [91], which promotes the growth of some tumors. CtIP contributes to transcriptional regulation of G1/S transition, DNA damage checkpoint signaling, and replication fork protection pathways. Therefore, SOX9 can be expected to be involved in these processes.

One significant difference between COLO357 and PANC-1 cells is the status of the *TP53* gene—a wild-type gene in COLO357 and a mutant gene in PANC-1. A mutation in the 273 codon of *TP53* affects the DNA-binding domain of the p53 protein, causing its normal interaction with target genes to be disrupted. In particular, the regulatory feedback loop between p53 and MDM-2 is disrupted [92].

Despite the presence of SOX9 recognition matrices in the promoter of the *TP53* gene, the Deseq2 program showed no significant change in *TP53* transcription levels in either of the two cell lines examined when *SOX9* was knocked down. This is not so surprising given that the underlying regulation of processes involving p53 is not related to changes in it’s gene transcription, but rather to stabilization of the protein for various reasons, including disruption of the MDM-2 feedback loop and interaction with specific microRNAs [93,94,95]. In Shih et al. [37], *TP53* was not identified as a gene that binds SOX9 factor in chromatin immunoprecipitation, although *TP53BP1*, *TP63*, and *TP73* bound SOX9. Nevertheless, we identified a number of p53-dependent processes affected by *SOX9* knockdown in COLO357 and PANC-1 cells (Table 8). We, therefore, hypothesize that SOX9 affects these processes indirectly through upstream or downstream p53 effectors, possibly complementing or modulating p53 activity.

In PANC-1 cells transfected with SOX9-specific siRNA, we detected lowered expression of *SOX12* and *SOX21* genes and of *SOX21* antisense RNA (*SOX21-AS*) relative to corresponding control cells. SOX12 was discovered as tumor progression and EMT promoting factor in breast and lung cancer, and as prognostic marker in lung cancer and renal cell carcinoma [1]. Evidence has been obtained that SOX12 may contribute to pancreatic cancer cell proliferation, migration, and invasion also through long noncoding and microRNAs mechanisms [96,97]. SOX21 plays a tumor suppressor role in central nervous system tumors as a complexing SOX1–3 inhibitor [5]. On the contrary, in pancreatic cancer, up-regulation of SOX21 by STAT6-activated *SOX21-AS* promotes cancer cells malignancy [98]. These are additional indications of tumor-suppressive ability of SOX9 in pancreatic cancer.

### 3.4. PDAssigner

Earlier we discovered that the down-regulation of SOX9 expression in the pancreatic cancer cell lines leads to cell line-specific changes in expression levels of epithelial and mesenchymal protein markers [16]. Thus, the suppression of SOX9 expression in COLO357–siSOX9 cells leads to a decrease in expression levels of some epithelial protein markers (CDH1 and some cytokeratins) in comparison with control COLO357–siNEG cells. At the same time, in PANC-1 cells, suppression of SOX9 expression did not result in decreased expression of epithelial markers. In COLO357–siSOX9 cells, the expression of the mesenchymal protein marker vimentin was reduced compared to control cells. A similar decrease was not detected in PANC-1 cells. Expression of another mesenchymal marker CDH2 protein was increased in PANC-1–siSOX9 cells compared to control PANC-1–siNEG cells. We conclude that the cellular effect of suppression of SOX9 protein expression may depend on the cell line-specific regulatory networks.

To investigate the possible effect of SOX9 knockdown on EMT, we determined expression levels of genes from the PDAssigner gene set [24], which, according to the authors, allows us to make a differentiation between epithelial and mesenchymal phenotypes of PAAD cells.

The results of the present work indicate that in COLO357 and PANC-1 cells, *SOX9* knockdown decreases the expression of genes of both classical PAAD phenotype and mesenchymal phenotype. There is no increase in expression of PDAssigner genes with the only exception of *CAV1* gene in PANC-1–siSOX9 cells. This confirms the conclusion made in our previous work that in the studied PAAD cells, *SOX9* knockdown does not lead to enhancement of the epithelial phenotype (status) in contrast to other tumors [16].

### 3.5. Transcription Factors

The study of transcription factor expression in COLO357 and PANC-1 cells shows that SOX9 knockdown affected the expression of many transcription factors. Thus, we found changes in the transcription levels of 26 transcription factors in COLO357–siSOX9 cells and 209 transcription factors in PANC-1–siSOX9 cells. This confirms the role of SOX9 as a master regulator of transcription. At the same time, only 10 transcription factors had altered expression in cells of both cell lines studied. Each of the identified transcription factors, in turn, has multiple targets. According to Database of Human Transcription Factors Targets [99], there are 5736 known targets for the AHR factor, 4255 targets for the E2F7 factor, and 11,283 targets for ZBTB42. Thus, SOX9 appears to be involved in the cascade regulation of a colossal number of biochemical processes and signaling pathways, which greatly complicates the analysis and interpretation of its functions. In general, SOX9 knockdown in COLO357 and PANC-1 cells activated transcription factors associated with the response to xenobiotics, suppression of differentiation and transcriptional regulation of pluripotent stem cells, and suppressed the activity of factors associated with circadian rhythms and cell cycle regulation, including those involving p53.

### 3.6. Genes Bound to SOX9 by ChIP

We examined the expression of 3730 genes for which SOX9 factor binding has been shown using ChIP [37] and identified 130 genes whose expression was altered in COLO357–siSOX9 cells and 418 genes with altered expression in PANC-1–siSOX9 cells relative to the corresponding control cells. Thus, dual evidence of regulation by SOX9 was obtained for these genes—SOX9 binding in immunoprecipitation and altered expression upon SOX9 knockdown. In cells of both cell lines, a total of 51 genes changed expression upon SOX9 knockdown, of which the expression of 11 genes increased and 35 genes decreased; the expression of 5 genes changed in different directions. Among the processes activated in cells of both cell lines, processes involving NGFR and the genes TAF13 and SGK1, regulators of p53 activity and transcription, were prominent; among those repressed, processes of signaling by interleukins and cell cycle regulation involving E2F7, CDC6, CKS1B, and other genes were prominent (Appendix A). This is consistent with the previously obtained data on the inhibition of cell proliferation during SOX9 knockdown [16].

### 3.7. Differentially Expressing Genes

We investigated the effect of SOX9 knockdown on the expression of approximately 1200 genes for which differential expression has been shown to occur in PAAD [29,38,39,40,41,42,43]. It turned out that both in the group of genes with increased expression in PAAD relative to the normal pancreas and in the group of genes with decreased expression in the tumor, the expression of part of genes is up-regulated and the other part of genes is down-regulated during SOX9 knockdown. We did not find statistically significant differences between these groups of genes. Thus, *SOX9* knockdown affects with approximately equal frequency both overexpressed and under-expressed genes in PAAD.

### 3.8. Prognostic Gene Signatures

One of the most important tasks of molecular oncology is to find diagnostic and prognostic markers of carcinogenesis for use in the clinic. A number of gene expression signatures associated with metastasis, survival, and prognosis of PAAD have been published [38,44,45,46,47,48,49,50,52,53,55]. SOX9 is also considered as a prognostic marker in a number of tumors, including PAAD [1]. In particular, it was reported that SOX9 expression can serve as a differentiating marker of pancreatic adenocarcinomas from solid pseudopapillary neoplasms [100], as well as a marker of pancreatic ductal lineage neoplasms [101], however, the authors did not find correlations of SOX9 expression with the grade of tumors.

We investigated changes in the expression of genes included in these signatures in COLO357 and PANC-1 cells upon SOX9 knockdown. Regularities could be detected only when analyzing genes proposed as markers of tumorigenicity and metastasis by Missiaglia et al. [38]. SOX9 knockdown in both studied cell lines reveals decreased expression of a group of genes whose increased expression is characteristic of primary tumors (cluster A) and genes whose expression is elevated in PAAD cell lines isolated from lymph node metastases (cluster C)—in particular, the *MEGF6* gene. In addition, increased expression of *SGK1* and *IRS1* genes was found in COLO357 and PANC-1 cells with suppressed SOX9. This may indicate a decrease in tumorigenic potential of these cells with simultaneous activation of intercellular interaction via molecular signals in cells with SOX9 knockdown.

For other signatures, we did not detect statistically significant changes in gene expression upon *SOX9* suppression. Perhaps some agreement of our results with the data of Missiaglia et al. [38] is due to the fact that both works were performed on cell line models, as well as to the fact that PANC-1 cells originated from the primary tumor, while COLO357 from a lymph node metastasis.

### 3.9. Drug Resistance Genes

One of the manifestations of oncogenic properties of SOX9 in cancer is the increase in tumor resistance to chemotherapy agents, in particular, to gemcitabine. According to Higashigara et al. [18], cells with high SOX9 expression, such as PANC-1, are more resistant to gemcitabine than cells with low expression of this gene, like BxPC-3. The authors attributed gemcitabine resistance to SOX9 ability to stimulate some stem cell traits in tumor cells. Thus, artificial suppression of SOX9 in PANC-1 cells decreased the number of cells carrying CD44 and CD24 markers, as well as decreased the ability of the cells to form spheres (sphere formation rate) and to initiate tumor in nude mice.

We have previously shown that the expression level of SOX9 in COLO357 and BxPC-3 cells is approximately equal but significantly lower than that in PANC-1 cells [16]. Accordingly, COLO357 cells are more sensitive to gemcitabine than PANC-1 cells [102,103].

We investigated the effect of SOX9 knockdown on the expression of genes associated with resistance or sensitivity of PAAD cells to antitumor drugs according to the literature—to gemcitabine, paclitaxel, SN38, 5-fluorouracil, and oxaliplatin [47,49,50,56,57,58,59]. COLO357–siSOX9 cells showed increased expression of a single *HIVEP2* gene associated with oxaliplatin resistance of PAAD organoids. Expression of other genes associated with both resistance and sensitivity to five antitumor drugs named above was down-regulated in COLO357–siSOX9 compared to control COLO357–siNEG cells. In PANC-1 cells, number of genes activated or repressed upon SOX9 knockdown did not differ significantly for genes associated with resistance or sensitivity to gemcitabine, paclitaxel, SN38, 5-fluorouracil, and oxaliplatin.

We analyzed specially expression of ATP-binding cassette (ABC) superfamily proteins. More than 40 of them function as membrane transporters, 20 of which are associated with multidrug resistance of tumors (MDR) [60]. In both COLO357 and PANC-1 cells with *SOX9* knockdown, only the *ABCB10* gene was suppressed relative to corresponding cells transfected with neutral siRNA. ABCB10 belongs to ABCB family of molecular transporters, several of which are associated with MDR [60]. A special function of ABCB10 is the export of an unknown physiological substrate from the mitochondrial matrix to the cytosol in an ATP-dependent manner [104] and participation in heme biosynthesis [105]. ABCB10 contributes to cisplatin resistance in epidermoid carcinoma cells [106]. Circ-RNA of ABCB10 may provide paclitaxel resistance of breast cancer cells [107] and cisplatin resistance of lung cancer cells [108]. There are no data on role of ABCB10 in antitumor drug response of pancreatic cancer. Thus, we found lowered expression of each drug response gene that was affected by *SOX9* knockdown in both cell lines. At the same time, increased expression of the xenobiotic response-related transcription factor gene *AHR* (R-HSA-211981) in Colo357 and PANC-1 cells with *SOX9* knockdown was detected.

In Colo357–siSOX9 cells, only the *ABCB1* gene was up-regulated compared to control COLO357 cells. ABCB1 is known also as MDR1 or P-glicoprotein. This protein translocates xenobiotics, drugs, phospholipids, and steroid hormones through the cell membrane, which may be a cause of resistance of cancer cells to more than 35 anticancer drugs including cisplatin, paclitaxel, doxorubicin, vincristine, and vinblastine [60]. In total, ABCB1 may have about 90 different substrates [109]. In PANC-1 with *SOX9* knockdown, we found decreased expression of *ABCA1* and *ABCA2* genes but increased expression of *ABCC5* and *ABCG* genes, all four genes being associated with multidrug resistance of tumors. Since the substrate sets of ABCA1-2, ABCB1, ABCC5, and ABCG2 overlap [60], the general effect of SOX9 suppression on overall or partial drug resistance of pancreatic cancer cells in context of these ABC proteins cannot be predicted with certainty.

Our data demonstrate ambiguous impact of SOX9 suppression on the expression of genes associated with antitumor drug resistance in pancreatic cancer cells. It must be emphasized that we studied only two cell lines, and other cell lines or tumors may respond to *SOX9* knockdown very differently. Nevertheless, we hope that the results obtained by us may stimulate deeper research on SOX9 role in drug resistance of cancer.

## 4. Materials and Methods

### 4.1. Materials and Cell Cultures

Chemicals were obtained from Sigma-Aldrich (Sigma-Aldrich, St. Louis, MO, USA), unless otherwise stated. Sera and cell culture media were obtained from Invitrogen (Invitrogen Corporation, Carlsbad, CA, USA). The list of primary and secondary antibodies used in the experiments is presented in Appendix A. The human pancreatic cancer cell PANC-1 line was obtained from the American Type Cell Collection (ATCC, Manassas, VA, USA) and COLO357 [22] cell culture was obtained from Dr. Klaus Felix (Heidelberg University Hospital, Heidelberg, Germany). PANC-1 cells were maintained in Dulbecco’s modified Eagle medium/Ham’s F12 (DMEM/F12, 1:1) containing 10% fetal calf serum, 2 mM L-glutamine, 100 U/mL penicillin, and 100 μg/mL streptomycin. COLO357 cells were maintained in RPMI 1640 medium containing 10% fetal calf serum, 2 mM L-glutamine, 55 nM 2-mercaptoethanol, 100 U/mL penicillin, and 100 μg/mL streptomycin. All cells were cultured at 37 °C in a humidified atmosphere with 95% air and 5% CO_2_.

### 4.2. Knockdown of SOX9 via RNA Interference

SiRNA#2 negative control (Cat #4390846, Lot #AS02F4T6, hereinafter, siNEG RNA) was purchased from Ambion, Inc. (Austin, TX, USA). Human SOX9 targeting siRNAs were synthesized by Syntol (Syntol JSC, Moscow, Russia). Sense sequences of the SOX9 siRNAs are presented in Appendix A. A total of 100 mcM sense and antisense oligoribonucleotides were mixed in equimolar concentrations in annealing buffer (50 mM Tris, pH 7.5, 100 mM NaCl) for final concentration of 20 mcM and annealed by heating for 2 min at 95C and gradual cooling to room temperature for 45 min. Three different *SOX9* siRNA duplexes were tested individually and in a combination. COLO357 cells (2 × 10^5^ cells/well) and PANC-1 cells (2 × 10^5^ cells/well) were transfected in 6-well plates with either 10 nM of control neutral siNEG or 10 nM pooled siSOX9 duplexes. SiRNA transfections were performed using the Lipofectamine RNAiMAX reagent (Invitrogen/Thermo Fisher Scientific, Waltham, MA USA), according to the manufacturer’s protocol. To achieve more effective transfection and the best depletion of the SOX9 protein, transfection was performed twice: 24 h after seeding the cells and 48 h after the first transfection. Twenty four hours after the second transfection, cells were harvested for further analysis [16].

### 4.3. Western Blot Analysis

Cell lysates were prepared from siSOX9-transfected test cells (PANC-1-siSOX9, COLO357-siSOX9) and siNEG-transfected control cells (PANC-1-siNEG, COLO357-siNEG), according to the previously described protocol [110]. Cell lysates were boiled in SDS sample buffer consisting of 1% SDS, 2% 2-mercaptoethanol, and 62 mM Tris-HCl, pH 6.8. Equal amounts of denatured protein lysates (20 μg of total protein) were subjected to SDS electrophoresis in 10–15% polyacrylamide gels, and then electrotransferred to an Immobilon-P membrane (Millipore, Bedford, MA, USA) using a Bio-Rad Trans-Blot SD cell (Bio-Rad Laboratories, Hercules, CA, USA). These membranes were then blocked with 5% milk in PBS-T (PBS containing 0.1% Tween 20) for 1 h at room temperature and incubated in PBS-T containing 5% milk and the relevant primary antibody overnight at 4 °C. After the final wash with PBS-T, the membranes were incubated in PBS-T containing goat anti-mouse or antirabbit antibody HRP conjugates (1:1000, Cell Signaling Technologies, Danvers, MA, USA) for 1 h. Membranes were washed with PBS-T, and signals were visualized with Clarity Western ECL solution (Bio-Rad Laboratories, Inc., Hercules, CA, USA) and Bio-Rad ChemiDoc Touch imaging device. Digital images of Western blot bands were quantified by densitometry using the Bio-Rad Image Lab software (version 5.2.1).

### 4.4. RNA-Sequencing (RNA-Seq)

Total RNA from each sample was extracted using the RNeasy Mini Kit (Qiagen GmbH, Hilden, Germany). The yield of RNA was first quantified by measuring total RNA concentration using a Qubit™ 2.0 fluorimeter (Thermo Fisher Scientific, Waltham, MA USA) with Qubit™ RNA HS Assay Kit (Thermo Fisher Scientific). The quality of the extracted RNA was estimated using the Agilent High Sensitivity RNA Screen Tape using the TapeStation 2200 (Agilent Technologies, Santa Clara, CA, USA). PolyA RNA isolation was performed with the NEBNext Poly(A) mRNA Magnetic Isolation Module (New England Biolabs, Ipswich, MA, USA). The NEBNext Ultra II Directional RNA Library Prep Kit for Illumina and the NEBNext Multiplex Oligos for Illumina (Index Primers Set 3, 4) (both from New England Biolabs, Ipswich, MA, USA) were used to prepare the libraries. Each library preparation was performed in three biological replicas for PANC-1, COLO357, PANC-1–siNeg, PANC-1–siSOX9, COLO357–siNeg, and COLO357–siSOX9. Finally, the resulting 12 individually indexed RNA-seq libraries were mixed in equimolar amounts, the final mixture was analyzed with Agilent TapeStation 2200 system. The median fragment length of the pooled fragments was 336 base pairs (bp), distributed between 200 and 700 bp. Sequencing was performed with the Illumina NovaSeq 6000 System (Illumina, San Diego, CA, USA). The numbers of reads are presented in Appendix A. The quality of the raw reads was assessed using the MultiQC tool (Galaxy Version 1.11 + galaxy0, accessed 25 May 2022 [111]). RNA-seq data were processed using the Galaxy online project [112]. Raw reads were mapped to a human reference genome (hg38) using HISAT2 (Galaxy Version 2.1.0 + galaxy 5, accessed on 25 May 2022, [113]). FeatureCounts (Galaxy Version 1.6.4 + galaxy 1, accessed on 25 May 2022, [114]) was used to count the reads of the annotated GENCODE genes, version 33, hg38 [114]. The resulting reads were converted to transcripts per kilobase million (TPM) values [115]. The DeSeq2 tool (Galaxy Version 2.11.40.8 + galaxy0, accessed 13 February 2024 [116]) was used to identify differentially expressed genes (DEG) in pairwise comparison. Expression was recognized as significant if TPM mean of three biological repeats was more than 5 and *p*-adj < 0.05. Fold change of expression more than 1.5 in siSOX9-transfected cells relative to siNEG-transfected cells (|log2(FC)| > 0.585) was considered as significant. Selected genes were validated by quantitative real-time PCR with primers presented in Appendix A.

### 4.5. Genes and Pathways Annotation

Functional classification of the genes revealed by RNA-seq was performed using Gene Ontology Resource [117,118], DAVID bioinformatics database [119]. Outdated gene names were converted to actual ones by g:Profiler (version: 13 February 2024 with Ensembl 111 and Ensembl genomes 57, [120]), SynGO (version: 12 December 2020, [121]), or HGNC Multi-symbol checker (accessed on 28 March 2024, [122]). Gene clasterization and expression heatmaps were constructed with Heatmapper [123] and Galaxy heatmap2 (Galaxy Release 24.0 + galaxy 1, usegalaxy.org, accessed on 15 March 2024, [114]). Gene annotations, pathway, and expression analyses were performed by means of Gene Ontology/Panther (GO Ontology database DOI: 10.5281/zenodo.10536401 released 17 January 2024, PANTHER overrepresentation test version 18.0 released 17 October 2023, accessed on 24 February 2024 [124]), Reactome (Pathway browser 3.7, database release 87, accessed on 26 February 2024, [125]) and GSEA-MSigDB (GSEA Desktop version 4.3.2, MSigDB version 2023.2 Hs, accessed on 22 February 2024, [27]) online databases. EMT genes lists were taken from dbEMT2 database (version 2.0, accessed on 14 December 2023, [34]). 

### 4.6. Statistical Analysis

Statistical significance was assessed using the Wilcoxon–Mann–Whitney or chi-square tests. For all the tests except noted especially, *p*-value < 0.05 was considered statistically significant. Calculations and graphical representations were performed using Microsoft ^®^ Excel 2016 (Microsoft Corp., Redmond, WA, USA) and Prism 8.4.3 (GraphPad Software, San Diego, CA, USA).

## 5. Conclusions

In the present work, we were the first to compare full transcriptomes of pancreatic cancer cell lines, one of which (PANC-1) originated from primary ductal adenocarcinoma and another (COLO357) isolated from a metastasis of pancreatic ductal adenocarcinoma to lymph node, under the condition of *SOX9* knockdown. Morphological and genetic traits of the cell lines were taken into account also.

Full transcriptome analysis of COLO357 and PANC-1 cells with *SOX9* knockdown show that the transcription factor SOX9, as a master regulator, is involved in the regulation of many genes and signaling pathways important in pancreatic development and carcinogenesis.

The activation of processes related to the transmission of various types of signals, differentiation, transcription, and methylation, with simultaneous suppression of the expression of genes involved in the regulation of the cell cycle and apoptosis, was observed as the most notable difference between the transcriptomes of COLO357 and PANC-1 cells with *SOX9* knockdown and control cells transfected with neutral siRNA. 

Among the genes that altered their expression upon SOX9 knockdown, we found transcription regulators, differentially expressed genes, transcription factors, oncogenes and tumor suppressor genes, genes of epithelial–mesenchymal transition, and genes responsible for antitumor drug resistance, as well as genes included in prognostic signatures for pancreatic cancer.

At the same time, a limitation of the work is that only two cell lines were used. Though the results obtained do not support using SOX9 as a diagnostic or prognostic marker in cancer, additional studies using other biological models are needed. Since the influence of SOX9 inhibition on the tumor cell transcriptomes is extremely diverse and often multidirectional, it may be more promising to focus on genes and factors with narrowly targeted specific functions in the search for new therapeutic and prognostic tumor markers.

## Figures and Tables

**Figure 1 ijms-26-02652-f001:**
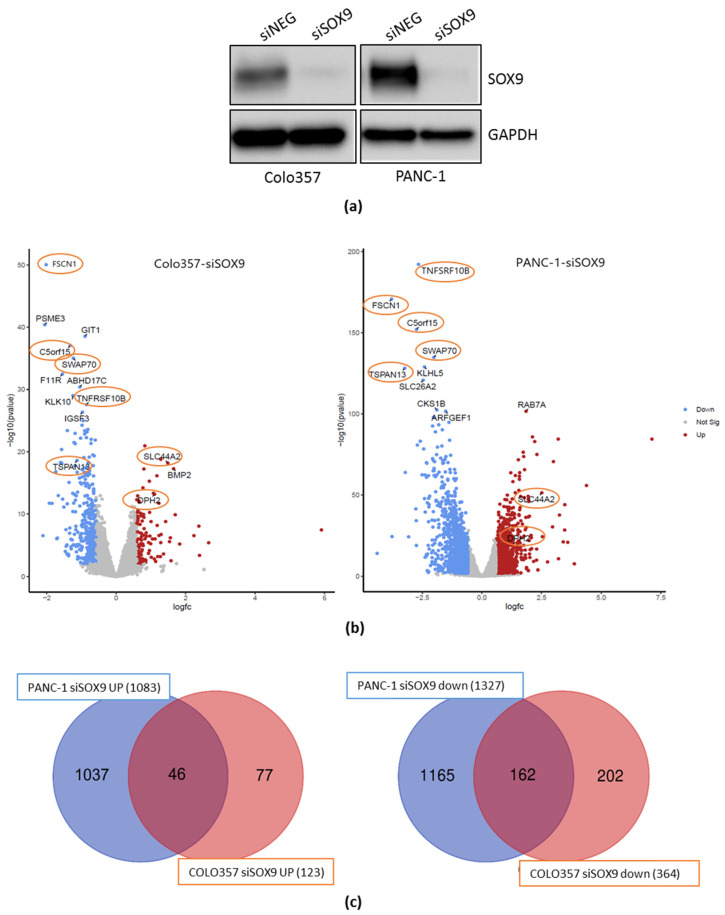
Effects of *SOX9* knockdown on transcriptomes of COLO357 and PANC-1 cells. (**a**) Western blot analysis of SOX9 protein expression in COLO357 and PANC-1 cells 72 h after transfection with *SOX9*-specific siRNA (siSOX9) or neutral siRNA (siNEG). GAPDH protein is used as a normalizing control. (**b**) Differential expression of genes (DEGs) in the pancreatic cancer cell lines Colo357 (**left**) and PANC-1 (**right**) after siRNA *SOX9* knockdown. Volcano plot displaying the distribution of the differentially expressed genes in siSOX9 cells samples with log2fold changes vs. −log10 (*p*-value). Top ten-fold change genes are denoted, some common genes are circled. Thresholds are fold change 1.5; *p*-adj < 0.05. (**c**) Venn diagram with overlapping sets of up-regulated (**left**) and down-regulated (**right**) genes in COLO357 and PANC-1 cells with *SOX9* knockdown lines relative to transcriptomes of corresponding cells transfected with neutral siRNA.

**Figure 2 ijms-26-02652-f002:**
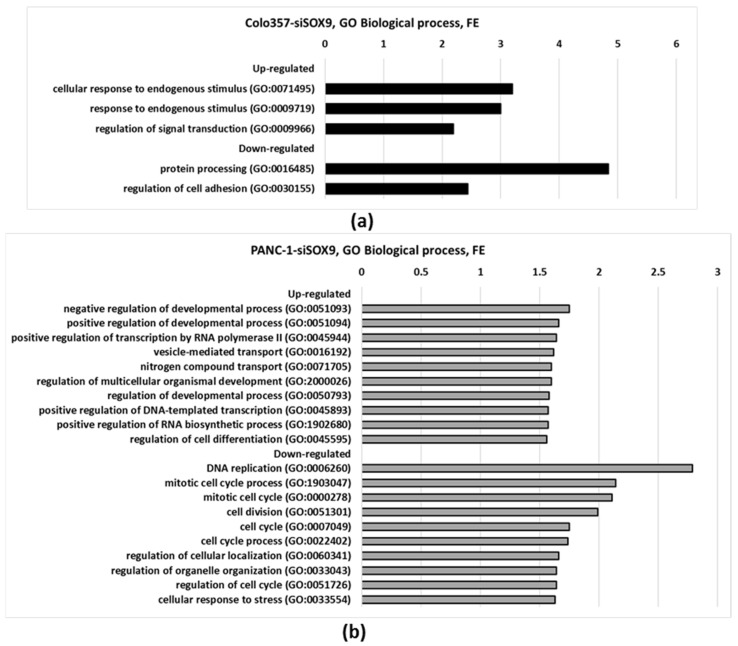
Gene ontology terms related to biological processes for gene expression data in pancreatic cancer cell lines with SOX9 knockdown compared to control cells of the same lines. (**a**) COLO357 cells. (**b**) PANC-1 cells (top ten by fold enrichment) Fisher test, *p*-adj < 0.05. FE—fold enrichment. See also Appendix A.

**Figure 3 ijms-26-02652-f003:**
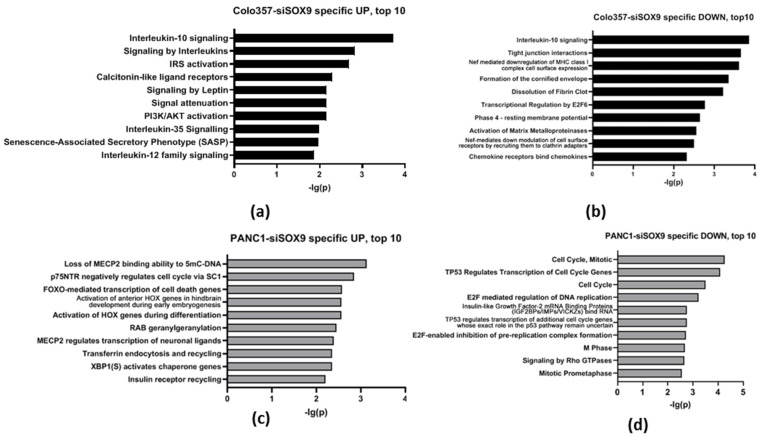
Enriched Reactome altered pathways of up-regulated genes (**a**) and down-regulated genes (**b**) in COLO357–siSOX9 cells, as well as up-regulated genes (**c**) and down-regulated genes (**d**) in PANC-1–siSOX9 cells relative to corresponding control cells. The top ten cell-specific altered pathways are represented. See also Table 4, Table 5 and Table 6 and Appendix A.

**Figure 4 ijms-26-02652-f004:**
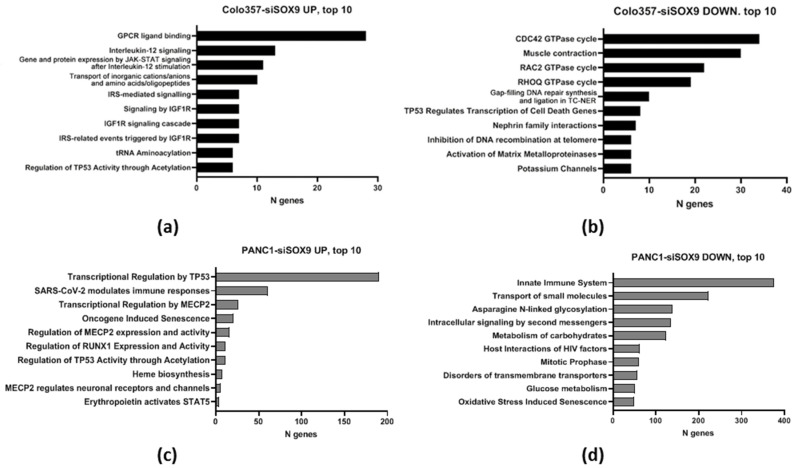
Top ten biological processes identified by the number of differentially expressed genes COLO357 (**a**,**b**) and PANC-1 (**c**,**d**) cells with SOX9 knockdown compared to control cells, as determined by Reactome-PADOG.

**Figure 5 ijms-26-02652-f005:**
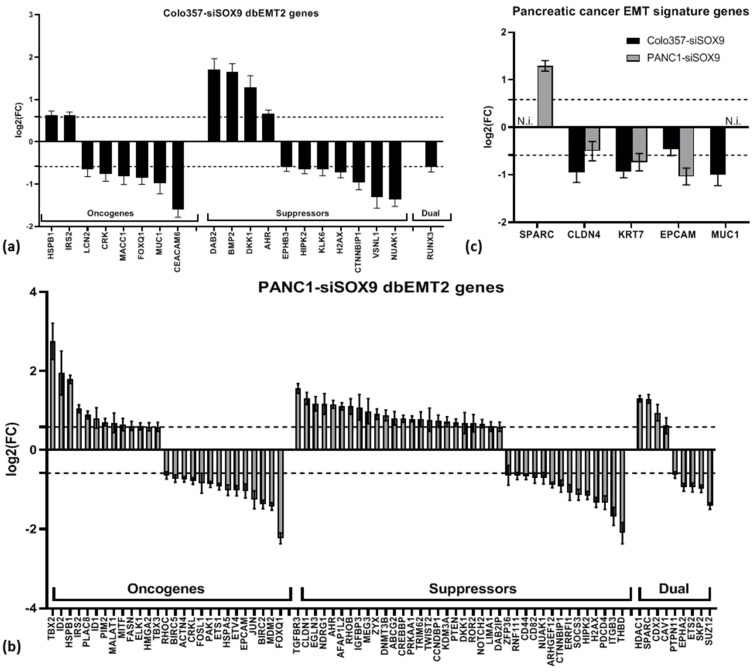
Alterations in epithelial–mesenchymal transition (EMT) genes from dbEMT2 database [34] in COLO357 and PANC-1 cells following *SOX9* knockdown compared to control cells. Genes were classified according to their roles in the EMT process as oncogenes, tumor suppressors, or dual-function genes; (**a**) COLO357–siSOX9; (**b**) PANC-1–siSOX9; (**c**) alterations of expression of pancreatic cancer EMT signature genes [35] in COLO357 and PANC-1 cells upon *SOX9* knockdown. Genes informative for any one of the two cell lines are presented. Dashed lines—threshold for fold change FC = 1.5. Log2(FC) is a characteristic of siSOX9/siNEG ratio for each cell line. See also Appendix A.

**Figure 6 ijms-26-02652-f006:**
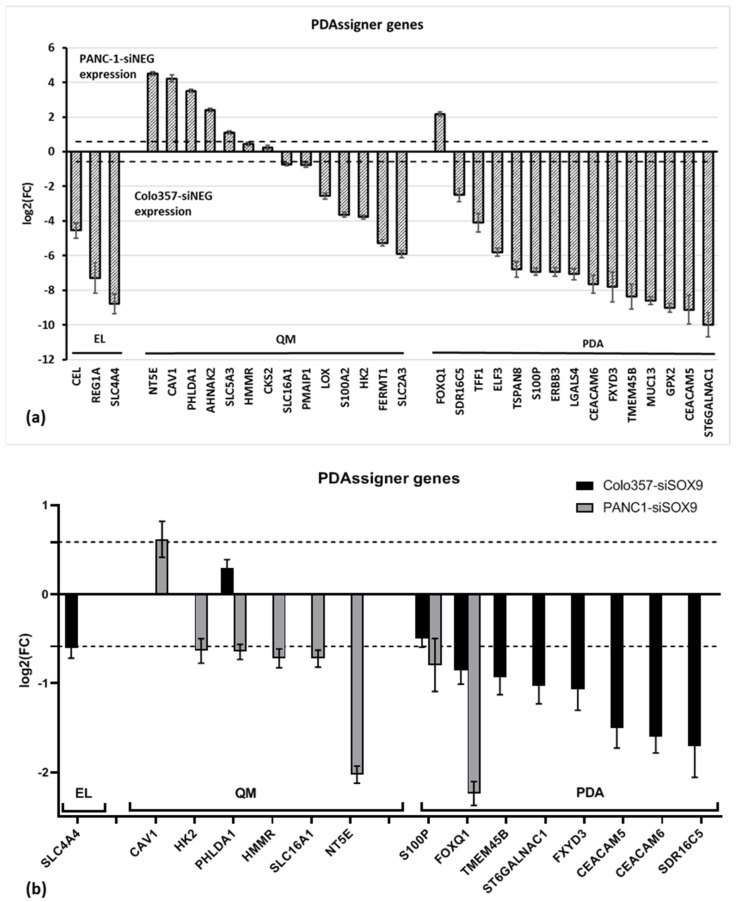
Expression of pancreatic cancer subtype specific genes according to PDAssigner [24] in PANC-1 versus COLO357 cells; (**a**) Comparison of cell lines both transfected by neutral (siNEG) siRNA, (**b**) comparison of COLO357 and PANC-1 cell lines transfected by SOX9-specific (siSOX9) siRNA relative to corresponding control cells transfected by siNEG RNA. Thresholds at fold change FC = 1.5 are shown by dashed lines. Genes with FC > 1.5 for any one cell line are presented. EL—exocrine-like PAAD subtype genes, QM—quasi-mesenchymal PAAD subtype genes, PDA—classic PAAD (epithelial) subtype genes. See also Appendix A.

**Figure 7 ijms-26-02652-f007:**
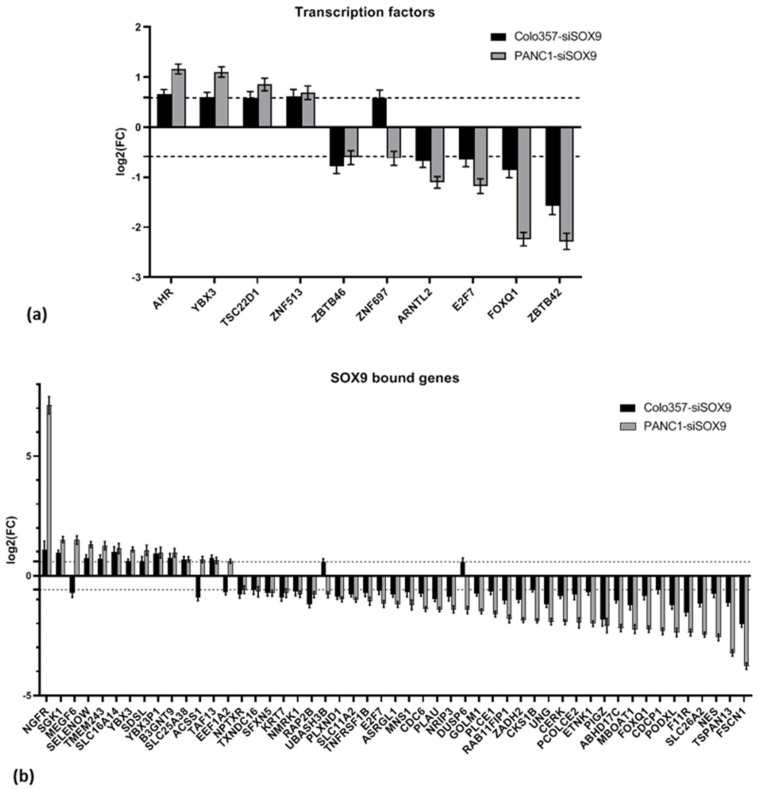
SOX9 as a master regulator of gene expression. (**a**) Transcription factor genes (from The Human Transcription Factors database, [36]) that exhibited changes in expression in COLO357–siSOX9 and PANC-1–siSOX9 cells. (**b**) Genes that bind SOX9 (based on chromatin immunoprecipitation data [37]) and exhibited changes in expression following SOX9 knockdown in COLO357 and PANC-1 cells (Section 2.2.9). Only genes with a fold change (FC) greater than 1.5 (dashed threshold) in both cell lines are shown.

**Figure 8 ijms-26-02652-f008:**
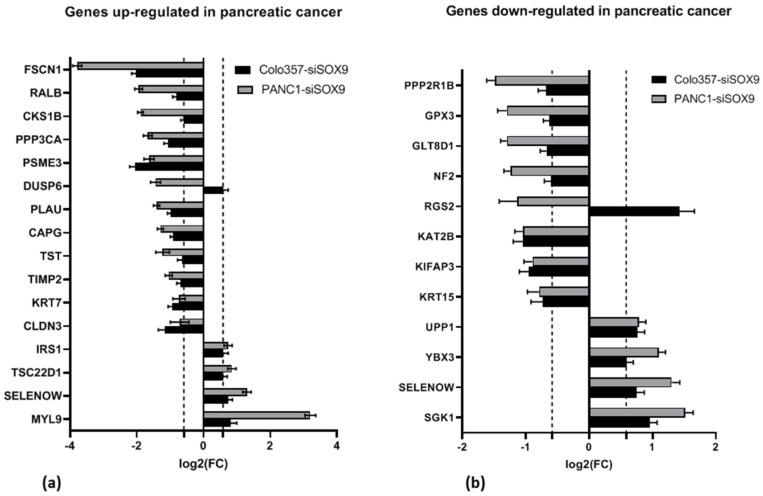
Comparison between the data obtained from clinical pancreatic cancer samples and cells (references are in Section 2.3.1) and SOX9 knockdown data. (**a**) Up-regulated genes in clinical samples and cells that show altered expression in COLO357–siSOX9 and PANC-1–siSOX9 cells relative to corresponding control cells. (**b**) The same, down-regulated genes. Genes with fold change FC greater than 1.5 (dotted lines) for both cell lines are presented only. See also Appendix A.

**Figure 10 ijms-26-02652-f010:**
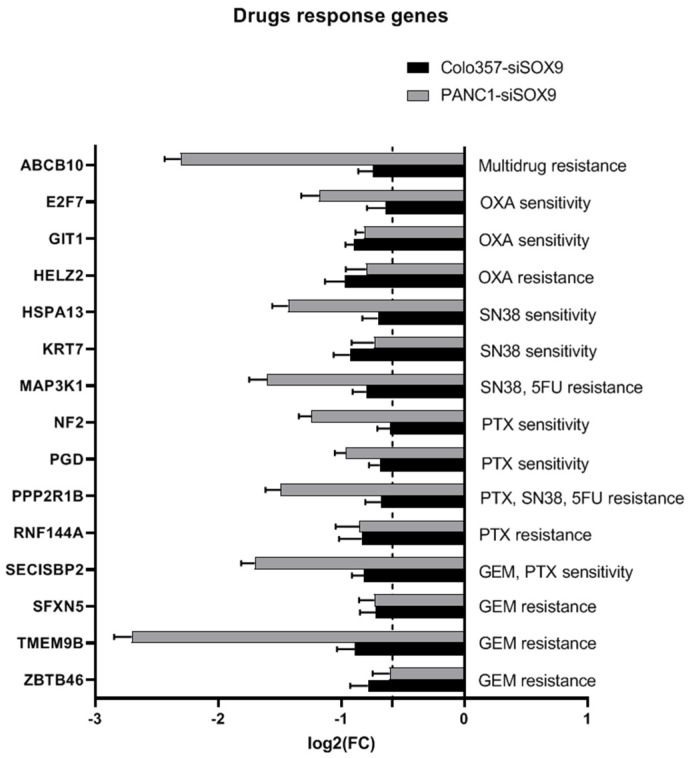
Alterations of the expression of genes associated with response to antitumor drugs (Section 2.3.3) in COLO357 and PANC-1 cells with *SOX9* knockdown relative to corresponding control cells Genes with fold change FC > 1.5 (dashed line) in both cell lines are shown only. See also Appendix A. OXA—oxaliplatin, SN38—topoisomerase I inhibitor SN38, 5FU—5-fluorouracil, PTX—paclitaxel, GEM—gemcitabine.

**Table 1 ijms-26-02652-t001:** Gene Ontology annotations of overrepresented transcripts in PAAD cells with SOX9 knock-down relative to cells transfected with neutral siRNA (common annotations only).

Gene Ontology	Up-Regulated Genes	Down-Regulated Genes
Biological process	Regulation of signal transduction (GO:0009966)	N.i. ^1^
Molecular function	N.i.	Protein binding (GO:0005515)
Cellular components	N.i.	Endomembrane system (GO:0012505)
Panther pathway	Gonadotropin-releasing hormone receptor pathway (P06664)	cytoplasm (GO:0005737).

^1^ N.i.—Not informative.

**Table 2 ijms-26-02652-t002:** Pathways affected by SOX9 knockdown in both COLO357 и PANC-1 cells by Reactome ^1^.

Pathway	COLO357–siSOX9 Genes	PANC-1–siSOX9 Genes
Common pathways with up-regulated genes		
R-HSA-112412 SOS-mediated signaling	***IRS1***, ***IRS2***	***IRS1***, ***IRS2***, *SOS1*
R-HSA-2892245 POU5F1 (OCT4), SOX2, NANOG repress genes related to differentiation	***TSC22D1***, ***DKK1***	*CDX2*, ***DKK1***, *EOMES*, ***TSC22D1***
R-HSA-9821993 replacement of protamines by nucleosomes in the male pronucleus	***H2BC12***, ***MYL9***	*H2BC8*, ***H2BC12***, *H2BC21*, *HDAC1*, *H3-3A*, ***MYL9***, *THBS1*, *TNRC6A*
R-HSA-171306 packaging of telomere ends R-HSA-73728 RNA polymerase I promoter opening R-HSA-5334118 DNA methylation	* **H2BC12** *	*DNMT3B*, ***H2BC12***, *H2BC8*, *H2BC21*, *H3-3A*
Common pathways with down-regulated genes		
R-HSA-69206 G1/S transition R-HSA-453279 mitotic G1 phase and G1/S transition	***CDC6***, ***CKS1B***, *PSMB10*, ***PSME3***, *PPP2R1B*, ***RBBP4***, *RRM2*	*CABLES1*, *CCN1*, *CCNB1*, ***CDC6***, *CDK1*, *CDK6*, ***CKS1B***, *MCM8*, *ORC1*, *ORC6*, *POLA2*, *POLE2*, *PPP2R1B*, *PRIM1*, *PRIM2*, *PSMA4*, *PSMD1*, ***PSME3***, ***RBBP4***, *RBL2*, *SKP2*
R-HSA-111464 SMAC(DIABLO)-mediated dissociation of IAP:caspase complexes R-HSA-111463 SMAC (DIABLO) binds to IAPs R-HSA-111469 SMAC, XIAP-regulated apoptotic response R-HSA-111459 activation of caspases through apoptosome-mediated cleavage	*APAF1*, ***CASP7***	***CASP7***, *DCP2*, *XIAP*

^1^ Pathways with Entities *p*-value < 0.05 are presented only. Genes of the same processes that are common for both COLO357 and PANC-1 cells are written in bold. See also Appendix A.

**Table 3 ijms-26-02652-t003:** Opposite altered pathways in PAAD cells with SOX9 knockdown relative to corresponding cells transfected with neutral siRNA.

Pathway	COLO357–siSOX9 Genes (Up-Regulated)	PANC-1–siSOX9 Genes (Down-Regulated)
R-HSA-2559582 senescence-associated secretory phenotype (SASP)	*H2BC12*, *IL6*	*CCN1*, *CDC26*, *CDC27*, *CDK6*, *CXCL8*, *FOS*, *H2AC11*, *H2AJ*, *H2AX*, *H2BC11*, *JUN*, *RPS6KA3*, *UBE2C*
R-HSA-2559583 cellular senescence	*H2BC12*, *IL6*, *MAP2K6*	*AGO4*, *CBX6*, *CBX8*, *CCN1*, *CDC26*, *CDC27*, *CDK6*, *CXCL8*, *ETS1*, *ETS2*, *FOS*, *H2AC11*, *H2AJ*, *H2AX*, *H2BC11*, *HMGA1*, *JUN*, *MAPK14*, *MAPK9*, *MDM2*, *MSN*, *POT1*, *RBBP4*, *RPS6KA3*, *SUZ12*, *TNRC6B*, *UBE2C*

**Table 4 ijms-26-02652-t004:** Pathways in COLO357–siSOX9 cells with up-regulated and down-regulated genes.

Pathway	COLO357–siSOX9 Genes (Up-Regulated)	COLO357-siSOX9 Genes (Down–Regulated)
R-HSA-6783783 interleukin-10 signaling	*CCL20*, *IL12A*, *IL6*	*CXCL1*, *CXCL2*, *CXCL3*, *IL1A*, *IL1R2*, *TNFRSF1B*
R-HSA-449147 signaling by interleukins	*CCL20*, *DUSP6*, *FYN*, *IL12A*, *IL6*, *IRS1*, *IRS2*, *LCP1*, *MAP2K6*, *ZEB1*	*CRK*, *CXCL1*, *CXCL2*, *CXCL3*, *FSCN1*, *IL1A*, *IL1R2*, *IL1RAP*, *LCN2*, *LGALS9*, *MUC1*, *PITPNA*, *PPP2R1B*, *PSMB10*, *PSME3*, *SERPINB2*, *TNFRSF1B*

**Table 7 ijms-26-02652-t007:** Oncogene sets expression change in PAAD cells with SOX9 knocked down ^1^.

Hallmark Name	SIZE	ES	NES	NOM *p*-Val	FDR q-Val	FWER *p*-Val
COLO357–siSOX9 up-regulated gene sets						
MYC_UP.V1_UP	30	0.42	2.24	0.0040	0.013	0.02
COLO357–siSOX9 down-regulated gene sets						
LEF1_UP.V1_DN	41	−0.55	−2.45	0	0	0
RPS14_DN.V1_UP	30	−0.54	−2.23	0	0.0050	0.009
PANC-1–siSOX9 up-regulated gene sets						
PTEN_DN.V1_UP	28	0.49	2.11	0.0035	0.013	0.012
TGFB_UP.V1_DN	65	0.37	2.03	0	0.014	0.026
PANC-1–siSOX9 down-regulated gene sets						
PRC2_EZH2_UP.V1_UP	62	−0.53	−2.36	0	0	0
CTIP_DN.V1_DN	25	−0.60	−2.14	0	0.0024	0.005

^1^ See comments to Table 6. Oncogene sets names and graphical data are presented in Appendix A.

**Table 8 ijms-26-02652-t008:** Processes dependent on p53 affected by SOX9 knock-down in PAAD cells ^1^.

Process	COLO357–siSOX9	PANC-1–siSOX9
R-HSA-6804754 regulation of TP53 expression (up)	N.i. ^2^	Appendix A
R-HSA-6804116 TP53 regulates transcription of genes involved in G1 cell cycle arrest (down)	Appendix A	Appendix A
R-HSA-6791312 TP53 regulates transcription of cell cycle genes (down)	Appendix A	Appendix A
R-HSA-6804115 TP53 regulates transcription of additional cell cycle genes whose exact role in the p53 pathway remain uncertain (down)	N.i.	Appendix A
R-HSA-6803211 TP53 regulates transcription of death receptors and ligands (down)	Appendix A	N.s. ^3^
R-HSA-5633008 TP53 regulates transcription of cell death genes	Appendix A (Down)	N.s.
R-HSA-6804758 regulation of TP53 activity through acetylation (up)	Appendix A	Appendix A
R-HSA-3700989 transcriptional regulation by TP53 (up)	N.i.	Appendix A
R-HSA-6804114 TP53 regulates transcription of genes involved in G2 cell cycle arrest (up)	N.s.	Appendix A
R-HSA-6804759 regulation of TP53 activity through association with co-factors (up)	N.s.	Appendix A
R-HSA-5633007 regulation of TP53 activity (up)	Appendix A	Appendix A
R-HSA-6804757 regulation of TP53 degradation (up)	Appendix A	Appendix A
R-HSA-6806003 regulation of TP53 expression and degradation (up)	Appendix A	Appendix A

^1^ Processes significant for any one line are presented only. ^2^ N.i.—Not identified. ^3^ N.s.—Not significant.

**Table 9 ijms-26-02652-t009:** Several important pathways with transcription factors differentially expressed in PAAD cells with SOX9 knock-down by Reactome.

Transcription Factor	Pathways ^1^
siSOX9 up-regulated TF’s	
AHR	Aryl hydrocarbon receptor signaling (R-HSA-8937144); Endogenous sterols (R-HSA-211976); Xenobiotics (R-HSA-211981)
TSC22D1	POU5F1 (OCT4), SOX2, NANOG repress genes related to differentiation (R-HSA-2892245); Transcriptional regulation of pluripotent stem cells (R-HSA-452723)
siSOX9 down-regulated TF’s	
ARNTL2	BMAL1:CLOCK,NPAS2 activates circadian gene expression (R-HSA-1368108)
E2F7	TP53 regulates transcription of genes involved in G1 cell cycle arrest (R-HSA-6804116) TP53 regulates transcription of cell cycle genes (R-HSA-6791312) Transcriptional regulation by TP53 (R-HSA-3700989)

^1^ Pathways from other sources are presented in Appendix A.

## Data Availability

The data presented in this study are available on Gene Expression Omnibus (GEO) database (Accession Number GSE287340, [126]).

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
