# Peer review of "Deciphering of SOX9 Functions in Pancreatic Cancer Cells"

_ijms, 2025, doi:10.3390/ijms26062652_

Round 1

Reviewer 1 Report

Comments and Suggestions for Authors

This article addresses the implications of SOX9 functions in pancreatic cancer cells. The topic is relevant, but major deficiencies identified in both content and form need to be addressed based on the specific recommendations below:

  1. The conclusion part of the abstract should be improved in terms of outcomes and the future research directions this study may refer to.
  2. L35-36 – In a scientific article, data should be presented rigorously, as should the bibliographic sources, so if ‘and others’ is mentioned, the source should be indicated.
  3. Multiple bibliographic citations (i.e. [2-5] etc.) can increase redundancy and decrease the correlation of the information presented with the references indicated. Either make the information more specific and divide the references, or better yet, remove some of them.
  4. The aim of the paper should be presented in the last paragraph of the introduction and should be addressed and detailed in terms of describing the contribution to the evaluated field and the elements of scientific novelty presented.
  5. L87, L112, L115, L136, L190, L193 …. L527, L529, L532 etc. - Reference source not found, a software implementation error appears in the bibliography. The issue must be resolved throughout the entire manuscript.
  6. The results section, according to rigorous scientific structure, involves only presenting the author’s own results, without bibliographic references being present. Any mention of technical matters/comparisons with other studies should be made in the discussion section.
  7. Examine the critical impact of SOX1 inactivation on inflammatory pathways, emphasizing how its disruption contributes to the activation of these pathways, and discuss the subsequent implications for increasing pancreatic cancer risk. It is imperative to understand this relationship to uncover potential therapeutic targets and provide valuable insights into the molecular mechanisms underlying pancreatic cancer development. I suggest checking and referring to: PMID: 37321055.
  8. It is advisable to discuss in summary also the potential impact of viral particles on SOX1 gene activity and whether chronic viral infections are associated with SOX1 inactivation. Assess if SOX1 inactivation is frequently observed in pancreatic cancer patients and if sequencing this gene could serve as a biomarker. Given SOX9’s role as a master regulator in pancreatic cancer, similar regulatory mechanisms may be at play with SOX1, linking its inactivation to tumor progression. Investigating these connections could provide valuable insights into the molecular drivers of pancreatic cancer and potential therapeutic targets. I suggest checking and referring to: PMID: 33114412 and PMID: 35055398.
  9. In the last paragraph of the Discussion section, it is advisable to detail the strengths, but especially the limitations of your study.
  10. L830 – This statement should be removed because the Materials and Methods section must include all the necessary data to ensure data reproducibility.
  11. L904, L906, section 4.6 etc. – The database should be included as a bibliographic reference in the form of a web page (with the link and access date as mandatory elements). This should be verified and applied throughout the entire manuscript wherever this instance occurs.

Author Response

    Quality of English Language

    (x) The English is fine and does not require any improvement.
    ( ) The English could be improved to more clearly express the research.

    This article addresses the implications of SOX9 functions in pancreatic cancer cells. The topic is relevant, but major deficiencies identified in both content and form need to be addressed based on the specific recommendations below:

    – We are grateful to the Reviewer for the helpful remarks, which undoubtedly helped to improve the quality of the article. Let us provide our responses in italic font indicating line numbers in the reviewed manuscript (with changes accepted):

    1. The conclusion part of the abstract should be improved in terms of outcomes and the future research directions this study may refer to.

    • The phrase “The results do not identify SOX9 as a prognostic marker in pancreatic cancer” has been changed to the phrase “Additional studies are needed to assess the properties and prognostic significance of SOX9 in pancreatic cancer using other biological models.” (Lines 25-27 of the revised manuscript with changes accepted). This much express the situation in concern of the subjects and results of the study more exactly.

    1. L35-36 – In a scientific article, data should be presented rigorously, as should the bibliographic sources, so if ‘and others’ is mentioned, the source should be indicated.
    • Only necessary references have been left in the first paragraph of the introduction (L. 38-41).
    1. Multiple bibliographic citations (i.e. [2-5] etc.) can increase redundancy and decrease the correlation of the information presented with the references indicated. Either make the information more specific and divide the references, or better yet, remove some of them.
    • Multiple citation have been divided; more detailed information on SOX9 biological functions has been added in the Introduction, i.e. on SOX9 pathways (L.46-55), SOX proteins interactions (L.56-67), inflammation (L.68-78).
    1. The aim of the paper should be presented in the last paragraph of the introduction and should be addressed and detailed in terms of describing the contribution to the evaluated field and the elements of scientific novelty presented.
    • The phrase “The objective of the present study was to enhance comprehension of the functions of SOX9 in the processes of proliferation, epithelial-mesenchymal transition, metastasis and other processes that are significant for the development of PAAD.” (L. 130-132) has been added in the last paragraph of the Introduction.
    • The phrase “In the present work, we were first to compare full transcriptomes of these cell lines under SOX9 knockdown taking into account the origin, morphological and genetic peculiarities of the cells” has been added into the Introduction (L. 113-115) and similar text was added as the first paragraph of the Conclusion (L.1074-1077).
    1. L87, L112, L115, L136, L190, L193 …. L527, L529, L532 etc. - Reference source not found, a software implementation error appears in the bibliography. The issue must be resolved throughout the entire manuscript.
    • These errors originated from internal WinWord references to in-text Figures and Tables during conversion of the manuscript to pdf format on IGMS site. To eliminate the issue we have changed these links to plain text all through the manuscript.
    1. The results section, according to rigorous scientific structure, involves only presenting the author’s own results, without bibliographic references being present. Any mention of technical matters/comparisons with other studies should be made in the discussion section.
    • The phrase “We have previously developed efficient siRNA duplexes…” with reference [6] (original lines 81-85) has been removed because all the necessary materials has been included in the manuscript under review and all the results described here are original. Other references in the Results section are essential because they are sources of gene sets (signatures) published elsewhere, expression of which we analyzed in the cells under SOX9 knockdown, and these references are a mandatory part of our Results.
    1. Examine the critical impact of SOX1 inactivation on inflammatory pathways, emphasizing how its disruption contributes to the activation of these pathways, and discuss the subsequent implications for increasing pancreatic cancer risk. It is imperative to understand this relationship to uncover potential therapeutic targets and provide valuable insights into the molecular mechanisms underlying pancreatic cancer development. I suggest checking and referring to: PMID: 37321055. – Nigam 2023
    • We wish to thank the reviewer for this very valuable comment. We included information on links of SOX9 with inflammatory and immunity environment in the Introduction (L. 68-78) and referred to the paper of Nigam, 2023 (ref. [7]). Though no significant impact of SOX9 knockdown on inflammation or immunity genes expression was revealed by GSEA, we found some meaningful correlation of by Reactome (Results, Table 4, Figure 3)) and discussed this evidence in the Discussion section (L. 706-715).

    1. It is advisable to discuss in summary also the potential impact of viral particles on SOX1 gene activity and whether chronic viral infections are associated with SOX1 inactivation. Assess if SOX1 inactivation is frequently observed in pancreatic cancer patients and if sequencing this gene could serve as a biomarker. Given SOX9’s role as a master regulator in pancreatic cancer, similar regulatory mechanisms may be at play with SOX1, linking its inactivation to tumor progression. Investigating these connections could provide valuable insights into the molecular drivers of pancreatic cancer and potential therapeutic targets. I suggest checking and referring to: PMID: 33114412 and PMID: 35055398.
    • The information on SOX proteins has been added into the Introduction (L. 56-67), Results (L. 340-355), Supplementary Table S12 and Discussion (L. 827-837). Alteration of SOX1 expression was not significant upon SOX9 knockdown in cell lines we studied.
    • The origin of cell lines we studied does not related to viral particles, so we consider viral particles beyond of the subject of our study.

    1. In the last paragraph of the Discussion section, it is advisable to detail the strengths, but especially the limitations of your study.
    • Special notes concerning the strength of our study have been added into Introduction  (L. 113-115) and Conclusion (L.1074-1078). The limitation notes of the study has been added into Discussion (L. 977-979) and Conclusion (L. 1093).
    1. L830 – This statement should be removed because the Materials and Methods section must include all the necessary data to ensure data reproducibility.
    • The statement has been removed; all necessary materials and methods are included in the manuscript.
    1. L904, L906, section 4.6 etc. – The database should be included as a bibliographic reference in the form of a web page (with the link and access date as mandatory elements). This should be verified and applied throughout the entire manuscript wherever this instance occurs.
    • The Internet links to the Databases has been removed because all the links are present in the papers we referred to when a Database is noted.

Reviewer 2 Report

Comments and Suggestions for Authors

In this original manuscript Nikitich Kashkin et al., investigated the functions of SOX9 in Pancreatic Cancer Cells. The authors' main objective was to reveal the relationship of SOX9 gene with other genes and processes characteristic of primary tumors and metastases using high-throughput full-transcriptome analysis. To this end, the authors performed full-transcriptome analysis of cells from two PAAD cell lines, PANC-1 and COLO357, under conditions of suppression of SOX9 gene expression using siRNAs. During the analyses, the authors found that suppression of SOX9 by small interfering RNAs (siRNAs) in PAAD cells results in decreased cell proliferation, inhibition of cell motility and migration activity, and induction of apoptosis. At the same time, it was observed that the effect of SOX9 on the expression of individual cell cycle regulatory proteins and markers of the epithelial-mesenchymal phenotype is ambiguous and depends on the specific cell line.

  • In my opinion, in the introduction section, the authors should describe in more detail the role of SOX9 in tumor cell biology. Although some sentences and references have been added to the introduction, more recent information should be included. This would facilitate reading and understanding of the article's objective for the readers, especially those who are not working in the field.
  • Regarding the methodology, the authors should add the original reference a the end of all the methodology used in the study, even if the methodology has already been described by the same group.
  • In the assays involving SOX9 silencing, please, explain in more detail how the entire transfection process was carried out. In my opinion, the described protocol is confusing.
  • Throughout the manuscript, there are several sentences "Error! Reference source not found.' Please, correct the information.
  • Item 2.2.9 needs to be revisited by the authors. The authors should add more information to facilitate reading and understanding by the readers.
  • In item 3.4, the authors describe: 'Previously, we have shown that SOX9 suppression decreases the expression of a number of epithelial markers (CDH1, cytokeratins 7, 18, and 19, and some proteins of intercellular contacts of epithelial cells) in the studied pancreatic cancer cells. At the same time, the expression levels of mesenchymal markers change differently in cells of different cell lines upon SOX9 suppression. In particular, in COLO357-siSOX9, the level of vimentin decreased but the level of SNAI2 increased relative to control COLO357-siNEG cells.' Please, explain the sentence better.
  • The figure legends need to be rewritten. The figures convey a lot of information, but the legends are lacking in content.
  • Regarding the data obtained on drug resistance, why do the authors not describe proteins belonging to the ABC superfamily, as well as proteins from the bcl-2 family, which are typically modulated in chemoresistant cells or those exhibiting the MDR phenotype? In many studies with different cell types, including pancreatic cancer cell lines, this has been demonstrated. Could the authors discuss this?
  • The English should be reviewed by a native English speaker.
  • The manuscript is interesting and addresses a topic of relevance in oncobiology. However, it needs to be better organized, and many sentences should be rewritten and modified to facilitate the reading of the scientific document. In particular, as mentioned above, the figure legends need more information.
Comments on the Quality of English Language

There are several typos throughout the manuscript. The English should be reviewed by a native English speaker.

Author Response

Reviewer 2

Open Review

Quality of English Language

( ) The English is fine and does not require any improvement.
(x) The English could be improved to more clearly express the research.

In this original manuscript Nikitich Kashkin et al., investigated the functions of SOX9 in Pancreatic Cancer Cells. The authors' main objective was to reveal the relationship of SOX9 gene with other genes and processes characteristic of primary tumors and metastases using high-throughput full-transcriptome analysis. To this end, the authors performed full-transcriptome analysis of cells from two PAAD cell lines, PANC-1 and COLO357, under conditions of suppression of SOX9 gene expression using siRNAs. During the analyses, the authors found that suppression of SOX9 by small interfering RNAs (siRNAs) in PAAD cells results in decreased cell proliferation, inhibition of cell motility and migration activity, and induction of apoptosis. At the same time, it was observed that the effect of SOX9 on the expression of individual cell cycle regulatory proteins and markers of the epithelial-mesenchymal phenotype is ambiguous and depends on the specific cell line.

– We would like to thank the Reviewer for very valuable comments and suggestions. We have also tried to implement all recommended changes. Please find our responses below in italic font with line numbers in the revised manuscript (with changes accepted):

  • In my opinion, in the introduction section, the authors should describe in more detail the role of SOX9 in tumor cell biology. Although some sentences and references have been added to the introduction, more recent information should be included. This would facilitate reading and understanding of the article's objective for the readers, especially those who are not working in the field.
  • The Introduction has been expanded in terms of biological functions of SOX9. I.e., intercourse of SOX9 with other member of SOX proteins (L. 56-67 of the revised manuscript) and interrelation  of SOX9 with inflammatory environment (L. 68-77) have been added.
  • Regarding the methodology, the authors should add the original reference a the end of all the methodology used in the study, even if the methodology has already been described by the same group.
  • The reference to our previous work has been relocated to the end of the section 4.2 (L. 1011).
  • In the assays involving SOX9 silencing, please, explain in more detail how the entire transfection process was carried out. In my opinion, the described protocol is confusing.
  • SOX9 silencing and transfection procedures have been explained in more details in Section 4.2 (L. 996-1011).
  • Throughout the manuscript, there are several sentences "Error! Reference source not found.' Please, correct the information.
  • These errors originated from WinWord internal links to Figures and Tables during conversion to pdf on the IGMS site. To resolve this issue we replaced all these links with plain text.
  • Item 2.2.9 needs to be revisited by the authors. The authors should add more information to facilitate reading and understanding by the readers.
  • Item 2.2.9 has been revisited; we hope it became more understandable.
  • In item 3.4, the authors describe: 'Previously, we have shown that SOX9 suppression decreases the expression of a number of epithelial markers (CDH1, cytokeratins 7, 18, and 19, and some proteins of intercellular contacts of epithelial cells) in the studied pancreatic cancer cells. At the same time, the expression levels of mesenchymal markers change differently in cells of different cell lines upon SOX9 suppression. In particular, in COLO357-siSOX9, the level of vimentin decreased but the level of SNAI2 increased relative to control COLO357-siNEG cells.' Please, explain the sentence better.
  • Item 3.4 has been revisited; we hope it became more understandable.
  • The figure legends need to be rewritten. The figures convey a lot of information, but the legends are lacking in content.
  • Figure legends in the manuscript have been expanded. If a contracted name of a pathway is present on a picture, then it is fully presented in a Supplementary table which is referred to in the Figure legend. Also, the legends for Supplementary figures S1-S4 have been transferred from service fields of the presentation to the slides and edited for more lucidity.
  • Regarding the data obtained on drug resistance, why do the authors not describe proteins belonging to the ABC superfamily, as well as proteins from the bcl-2 family, which are typically modulated in chemoresistant cells or those exhibiting the MDR phenotype? In many studies with different cell types, including pancreatic cancer cell lines, this has been demonstrated. Could the authors discuss this?
  • The ABC superfamily genes as well as other genes of cancer drug resistance are included in GSEA gene sets as well as in GeneOntology annotations and Reactome pathways. We did not find correlations for these genes characteristic to pancreatic cancer by analysis of databases named so we did not mentioned these gene sets in the manuscript. Nevertheless, according to the suggestion of the Reviewer, we analyzed effect of SOX9 knockdown on ABC genes expression and present the results now in section 2.3.3 (Results, L. 539-545), 3.9 (Discussion, L. 950-964) and Supplementary table S19.
  • The English should be reviewed by a native English speaker.
  • The English has been revisited thoroughly.

Round 2

Reviewer 1 Report

Comments and Suggestions for Authors

The authors improved their manuscript

Comments on the Quality of English Language

Resonable English

Reviewer 2 Report

Comments and Suggestions for Authors

I thank the authors for their efforts to improve the quality of the manuscript.